# DIMENSION-FREE DECISION CALIBRATION FOR NON-LINEAR LOSS FUNCTIONS

**Jingwu Tang, Jiayun Wu, Zhiwei Steven Wu, Jiahao Zhang** [*]

School of Computer Science

Carnegie Mellon University

Pittsburgh, PA 15213, USA

`{jingwut,jiayunw,zstevenwu,jiahaoz4}@cmu.edu`

## ABSTRACT

When model predictions inform downstream decisions, a natural question is under what conditions can the decision-makers simply respond to the predictions as if they were the true outcomes. The recently proposed notion of decision calibration (Zhao et al., 2021) addresses this by requiring predictions to be unbiased conditional on the best-response actions induced by the predictions. This relaxation of classical calibration avoids the exponential sample complexity in high-dimensional outcome spaces. However, existing guarantees are limited to linear losses. A natural strategy for nonlinear losses is to embed outcomes $y$ into an $m$-dimensional feature space $\phi(y)$ and approximate losses linearly in $\phi(y)$. Yet even simple nonlinear functions can demand exponentially large or infinite feature dimensions, raising the open question of whether decision calibration can be achieved with complexity independent of the feature dimension $m$. We begin with a negative result: even verifying decision calibration under standard deterministic best response inherently requires sample complexity polynomial in $m$. To overcome this barrier, we study a smooth variant where agents follow quantal responses. This smooth relaxation admits dimension-free algorithms: given $\mathrm{poly}(|\mathcal{A}|, 1/\epsilon)$ samples and any initial predictor $p$, our introducded algorithm efficiently test and achieve decision calibration for broad function classes which can be well-approximated by bounded-norm functions in (possibly infinite-dimensional) separable RKHS, including piecewise linear, Cobb–Douglas loss functions, and any Lipschitz differentiable functions.

## 1 INTRODUCTION

Machine learning models increasingly underpin decisions in high-stakes scenarios, such as medical diagnosis and financial forecasting. In these settings, model predictions inform downstream decision-makers who act to optimize their utilities. Formally, there is an underlying distribution $\mathcal{D}$ over the spaces of covariates $\mathcal{X}$ and outcomes $\mathcal{Y}$, and the goal is to learn a predictor $p \colon \mathcal{X} \to \mathcal{Y}$ that supports decision-making. Given an action set $\mathcal{A}$, a decision-maker follows a decision rule $k \colon \mathcal{X} \to \Delta(\mathcal{A})$ that utilizes predictions to minimize the expected loss incurred by a loss function $\ell \colon \mathcal{A} \times \mathcal{Y} \to \mathbb{R}$. Often, these scenarios encompass not just a single, known loss function but rather a broad class of potential loss functions $\mathcal{L}$. For instance, different stakeholders in healthcare might prioritize different aspects of the outcome, while financial investors vary in their tolerance for risk.

*When should a decision-maker treat a prediction $p(x)$ as if it were the true outcome $y$?* Calibration provides a principled answer. A predictor is calibrated if, conditioned on every output value $v$, the true outcome is on average equal to $v$. That is, $\mathbb{E}[Y \mid p(X) = v] = v$.

While calibration can be trivially achieved with a constant predictor $p(x) = \mathbb{E}[Y]$, such predictors are uninformative. In practice, calibration is often enforced via post-processing to simultaneously recalibrate the model and reduce its mean square error of prediction. If the loss function $\ell(a, y)$

---

[*]All authors are listed in alphabetical order.

is linear in $y$, decision-makers can treat predictions as reliable substitutes for outcomes. Best responses computed from calibrated predictions are indeed optimal actions for true outcomes, given the information from $p(X)$ (Foster & Vohra, 1999; Noarov et al., 2023). However, achieving calibration in high-dimensional outcome spaces is computationally and statistically intractable, because it requires exponentially many samples to verify the unbiasedness of prediction over exponentially many events of $\{p(x) = v\}$ (Gopalan et al., 2024a).

To circumvent this curse of dimensionality, Zhao et al. (2021) introduced the weaker notion of *decision calibration*. Unbiased predictions are only required for events relevant to action selection. As a result, decision calibration ensures that predictions $p(x)$ yield loss estimates that are statistically indistinguishable from the true losses from the perspective of decision-makers. For a loss function $\ell \in \mathcal{L}$ and a decision rule $k$, a predictor $p$ is decision calibrated if

$$\mathbb{E}_{(x,y)\sim\mathcal{D}}\mathbb{E}_{a\sim k(x)}[\ell(a, y)] = \mathbb{E}_{(x,y)\sim\mathcal{D}}\mathbb{E}_{a\sim k(x)}[\ell(a, p(x))].$$

Notably, both the verification and the post-processing algorithm for decision calibration cost polynomial time under high-dimensional settings. We defer additional discussion of related work to Appendix E.

However, prior work on calibration for decision-making (e.g., Foster & Vohra (1999); Zhao et al. (2021); Noarov et al. (2023)) have focused primarily on linear loss functions. Linearity is crucial for optimality of best-response actions to calibrated predictions. Yet, many real-world decision-making scenarios naturally involve non-linearities. For example, risk-averse investors penalize large losses more heavily. Clinicians have risk-sensitive objectives that assign greater weight to severe medical outcomes.

A natural strategy is to linearize loss functions through a feature expansion $\phi : \mathcal{Y} \to \mathcal{H}$, such that the loss function $\ell(a, y)$ can be expressed (or well approximated) as a linear operator in the higher-dimensional space $\mathcal{H}$:

$$\ell(a, y) = \langle r_\ell(a), \phi(y)\rangle_\mathcal{H}.$$

In this paper, we learn a decision calibrated predictor $p : \mathcal{X} \to \mathcal{H}$ such that $p(x)$ induces loss estimates indistinguishable from those of $\phi(y)$. However, practical loss function classes—such as certain subclasses of Lipschitz functions—typically require exponentially large dimensions of feature expansion. Since prior results on decision calibration (e.g., Zhao et al. (2021)) have sample complexity polynomial in the outcome dimension, standard decision calibration becomes computationally and statistically intractable when applied to these high-dimensional feature expansions. To address this challenge, we develop the framework of *dimension-free decision calibration*, which extends decision calibration to broad classes of non-linear loss functions without incurring dependence of sample complexity on the dimension of the feature space $\mathcal{H}$.

**Our contributions.**

1. **Lower bound under deterministic best response.** The standard "hard-max" best response decision rule selects the action which minimizes the loss assuming the outcome is perfectly predicted. For a $m$-dimensional feature space $\mathcal{H}$, we show that auditing decision calibration requires $\Omega(\sqrt{m})$ samples. The lower bound is proved by constructing two nearly indistinguishable distributions—one decision calibrated and the other miscalibrated—using the fact that there exist $d$ points that can be shattered by halfspaces in a $d$-dimensional space. To the best of our knowledge, this is the first lower bound of statistical complexity established for decision calibration. Our lower bound provides evidence that non-trivial dimension-free decision calibration algorithms do not exist under the deterministic best response decision rule, motivating our adoption of a smooth decision rule.

2. **Dimension-free auditing under smooth best response.** We then consider quantal response, a smooth optimal decision rule that stochastically select actions according to loss estimates. Quantal response has been extensively studied in economics and decision theory (McFadden et al., 1976; McKelvey & Palfrey, 1995), as it naturally captures bounded rationality and accounts for probabilistic decision-making.

$$\tilde{k}_{p,\ell}(x, a) = \frac{e^{-\beta\langle r_\ell(a), p(x)\rangle_\mathcal{H}}}{\sum_{a'} e^{-\beta\langle r_\ell(a'), p(x)\rangle_\mathcal{H}}}.$$

For quantal responses, we show that there exists a *dimension-free auditing algorithm*. With high probability, it can identify a loss function witnessing an $\epsilon/2$-decision calibration error whenever

the predictor has $\epsilon$-decision calibration error, using only $\mathrm{poly}(|\mathcal{A}|, 1/\epsilon, \beta)$ samples, independent of $m$. This sharp dimension-free guarantee is achieved through a carefully designed pseudometric that projects loss vectors in high-dimensional space into one-dimensional space. By exploiting the smoothness of the quantal response, the covering number under this pseudometric remains bounded, whereas the covering number under the standard metric would be infinite when $m$ is unbounded.

3. **Dimension-free algorithm for decision calibration.** Building on this auditing tool, we design a patching procedure that post-processes any predictor into one that is $\epsilon$-decision calibrated, without worsening its mean square error. This guarantee applies broadly to function classes representable or well-approximated by bounded-norm functions in an RKHS, including piecewise linear, Cobb–Douglas functions, and more generally, any Lipschitz differentiable functions. Notably, in terms of $\epsilon$-dependence, our algorithm improves upon the finite-sample version of the algorithm in Zhao et al. (2021). Our sample complexity scales as $1/\epsilon^4$, compared to $1/\epsilon^6$ in their algorithm.

## 2 PRELIMINARIES

**Notations** We consider the prediction problem for decision making with a context space $\mathcal{X}$ and a compact convex outcome space $\mathcal{Y} \subseteq \mathbb{R}^d$. Let $\mathcal{D}$ denote the distribution over $\mathcal{X} \times \mathcal{Y}$. Given any dataset $D = \{(x_i, y_i)\}_{i=1}^n$ that is drawn i.i.d from $\mathcal{D}$ and any function $\psi : \mathcal{X} \times \mathcal{Y} \to \mathbb{R}$, define the empirical expectation as $\hat{\mathbb{E}}_D[\psi(x, y)] = \frac{1}{n} \sum_{i=1}^n \psi(x_i, y_i)$. For any integer $n$, we use $[n]$ to denote the class $\{1, \cdots, n\}$.

### 2.1 LOSS FUNCTIONS AND UNIFORM APPROXIMATIONS

We model downstream decision makers as having a finite action space $\mathcal{A}$ and a loss function $\ell : \mathcal{A} \times \mathcal{Y} \to [0, 1]$, which maps an action-outcome pair to a bounded loss. Let $\mathcal{L}$ denote a family of such loss functions. To handle *nonlinear* losses, we adopt a standard approach of approximating them via a feature mapping $\phi : \mathcal{Y} \to \mathcal{H}$, where $\mathcal{H}$ is a (possibly infinite-dimensional) feature space. The idea is to approximate each $\ell \in \mathcal{L}$ by a linear function of $\phi(y)$. Once this approximation is established, we show in the following sections that decision calibration becomes achievable for such loss classes. When the feature space is finite-dimensional, we write $\mathcal{H} = \mathbb{R}^m$ with $\dim(\mathcal{H}) = m < \infty$. We also consider the case where $\mathcal{H}$ is a separable reproducing kernel Hilbert space (RKHS), which has a countable orthonormal basis, and $\dim(\mathcal{H})$ can be $\infty$.

We formally define this approximation framework as follows:

**Definition 2.1.** *Let $\phi : \mathcal{Y} \to \mathcal{H}$ be a feature map and $\mathcal{L}$ a family of loss functions. We say that $\phi$ provides a $(\dim(\mathcal{H}), \lambda, \epsilon)$-uniform approximation to $\mathcal{L}$ if for every $\ell \in \mathcal{L}$, there exists a function $r_\ell : \mathcal{A} \to \mathcal{H}$ such that $|\langle r_\ell(a), \phi(y) \rangle_{\mathcal{H}} - \ell(a, y)| \leq \epsilon$ and $\|r_\ell(a)\|_{\mathcal{H}} \leq \lambda$ for all $a \in \mathcal{A}$ and $y \in \mathcal{Y}$.*

Intuitively, Definition 2.1 requires the function $\ell(a, \cdot) : \mathcal{Y} \to \mathbb{R}$ to be uniformly approximated by functions $g_a : \mathcal{Y} \to \mathbb{R}$ that are linear in some feature space for any $a$. We provide two families of functions from the economics literature as examples that are linear in an infinite-dimensional feature space in Appendix F.

### 2.2 PREDICTORS AND LOSS ESTIMATORS

We now define the notion of a predictor given a feature mapping $\phi : \mathcal{Y} \to \mathcal{H}$. A *predictor* is a function $p : \mathcal{X} \to \mathcal{H}$, interpreted as estimating the conditional expectation $\mathbb{E}[\phi(y) \mid x]$. Since the feature space $\mathcal{H}$ can be high-dimensional or even infinite-dimensional, the predictor $p(x)$ can be complex and may lack an intuitive interpretation for downstream decision makers.

To address this, we do not expose the predictor directly. Instead, we use it to construct a *loss estimator* $f_p$, which takes as input a context $x$, an action $a$, and a loss function $\ell$, and outputs an estimate of the expected loss $\ell(a, y)$ given $x$. We formalize this notion below:

**Definition 2.2** (Loss Estimator). *A loss estimator is a function $f : \mathcal{X} \times \mathcal{A} \times \mathcal{L} \to \mathbb{R}$. For any context $x \in \mathcal{X}$, action $a \in \mathcal{A}$, and loss function $\ell \in \mathcal{L}$, the output $f(x, a, \ell)$ estimates the expected loss $\mathbb{E}[\ell(a, y) \mid x]$.*

Although the definition of $f$ does not require an explicit association with a predictor, in our approach the learned loss estimator is *implicitly* derived from an underlying predictor $p$. Specifically, when such a predictor is maintained, the loss estimator takes the form $f_p(x, a, \ell) = \langle r_\ell(a), p(x) \rangle$, where $r_\ell(a)$ is the coefficient vector associated with the loss function $\ell$, as defined previously.

## 2.3 DECISION RULES AND DECISION CALIBRATION

In an ideal setting, if a decision maker with loss function $\ell$ has access to the full distribution $\mathcal{D}$, they can compute and play the optimal action: $a^* = \arg\min_{a \in \mathcal{A}} \mathbb{E}_\mathcal{D}[\ell(a, y)]$. However, in practice, decision makers do not have access to the full distribution. Instead, they rely on the loss estimator $f$ to make decisions. Given a context $x$, the decision maker queries the estimated loss $f(x, a, \ell)$ for each action $a \in \mathcal{A}$ and selects an action accordingly.

We formalize the decision maker's behavior via a *decision rule*, which is a function $k : \mathcal{X} \times \mathcal{A} \to [0, 1]$, representing the probability of selecting action $a$ given context $x$. A common strategy is to select the action that minimizes the estimated expected loss:

**Definition 2.3** (Optimal Decision Rule). *For a given loss function $\ell$ and loss estimator $f$, the optimal decision rule is defined as:*

$$k_{f,\ell}(x, a) = \begin{cases} 1 & \text{if } a = \arg\min_{a' \in \mathcal{A}} f(x, a', \ell), \\ 0 & \text{otherwise.} \end{cases}$$

We also consider a *smoothed* version of the optimal decision rule, commonly referred to as the *quantal response* model in economics and decision theory. The quantal response model has been extensively studied in the literature (see e.g., (McFadden et al., 1976; McKelvey & Palfrey, 1995)).

**Definition 2.4** (Smooth Optimal Decision Rule). *For a loss function $\ell$, loss estimator $f$, and inverse-temperature parameter $\beta > 0$, the smoothed optimal decision rule is defined as:*

$$\tilde{k}_{f,\ell}(x, a) = \frac{e^{-\beta f(x, a, \ell)}}{\sum_{a'} e^{-\beta f(x, a', \ell)}}.$$

For convenience, we sometimes use $k(x)$ to denote the probability distribution over actions induced by a decision rule $k$. We now restate the definition of *decision calibration*, originally introduced by Zhao et al. (2021), with our notion of the loss estimator:

**Definition 2.5** (Decision Calibration). *Let $\mathcal{L}$ be a class of loss functions and $\mathcal{K}$ be a class of decision rules. A loss estimator $f$ is said to be $(\mathcal{L}, \mathcal{K})$-decision calibrated if for every $\ell \in \mathcal{L}$ and every decision rule $k \in \mathcal{K}$,*

$$\mathbb{E}_{(x,y) \sim \mathcal{D}} \mathbb{E}_{a \sim k(x)}[\ell(a, y)] = \mathbb{E}_{(x,y) \sim \mathcal{D}} \mathbb{E}_{a \sim k(x)}[f(x, a, \ell)]. \tag{1}$$

*We define the decision calibration error as:*

$$\text{decCE}_{\mathcal{L}, \mathcal{K}}(f) := \sup_{\ell \in \mathcal{L}, \, k \in \mathcal{K}} \left| \mathbb{E}_{(x,y) \sim \mathcal{D}} \mathbb{E}_{a \sim k(x)}[\ell(a, y)] - \mathbb{E}_{(x,y) \sim \mathcal{D}} \mathbb{E}_{a \sim k(x)}[f(x, a, \ell)] \right|.$$

*We say that a loss estimator is $(\mathcal{L}, \mathcal{K}, \epsilon)$-decision calibrated if $\text{decCE}_{\mathcal{L}, \mathcal{K}}(f) \leq \epsilon$.*

To interpret equation 1, the left-hand side represents the *true expected loss* incurred when the agent follows the decision rule $k$, while the right-hand side represents the *estimated expected loss* based solely on the loss estimator $f$. The agent can compute this estimate without access to the true outcome $y$. Intuitively, decision calibration ensures that the estimator $f$ is accurate across all relevant loss functions and decision rules.

We use $\mathcal{K}_\mathcal{L} := \{k_\ell | \ell \in \mathcal{L}\}$ to denote the class of decision rules induced by any loss function $\ell \in \mathcal{L}$ under the best response decision rule. Similarly, we use $\tilde{\mathcal{K}}_{\mathcal{L}_\mathcal{H}} := \{\tilde{k}_\ell | \ell \in \mathcal{L}\}$ to denote the class of decision rules induced by any loss function $\ell \in \mathcal{L}$ under the smooth best response decision rule.

We now discuss how the uniform approximation can help to achieve decision calibration. Let $\mathcal{L}_\phi$ denote the class of loss functions for which the feature mapping $\phi : \mathcal{Y} \to \mathcal{H}$ gives $(\dim(\mathcal{H}), \lambda, \frac{\epsilon}{2})$-uniform approximations and let $\hat{\mathcal{L}}_\phi = \{\hat{\ell} : \hat{\ell}(a, y) = r_\ell(a) \cdot \phi(y)\}$ denote the associated class of linear functions. Then given any predictor $p : \mathcal{X} \to \mathcal{H}$, we can define the loss estimator $f_p$ as

$$f_p(x, a, l) = \langle r_\ell(a), p(x) \rangle_\mathcal{H}$$

for any context $x \in \mathcal{X}$, action $a \in \mathcal{A}$ and loss function $\ell \in \mathcal{L}_\phi$. The following lemma shows that if the loss estimator $f_p$ is $\epsilon/2$-decision calibrated for class $\hat{\mathcal{L}}_\phi$, it is $\epsilon$-decision calibrated for class $\mathcal{L}_\phi$.

**Lemma 2.1.** *Let $\mathcal{L}_\phi$ denote the class of loss functions for which the feature mapping $\phi : \mathcal{Y} \to \mathcal{H}$ gives $(\dim(\mathcal{H}), \lambda, \frac{\epsilon}{2})$-uniform approximations and let $\hat{\mathcal{L}}_\phi = \{\hat{\ell} : \hat{\ell}(a, y) = r_\ell(a) \cdot \phi(y)\}$ denote the associated class of linear functions. For any predictor $p : \mathcal{X} \to \mathcal{H}$, any class of decision rule $\mathcal{K}$ and $\epsilon > 0$, if the loss estimator $f_p$ is $(\hat{\mathcal{L}}_\phi, \mathcal{K}, \epsilon/2)$-decision calibrated, then $f_p$ is $(\mathcal{L}_\phi, \mathcal{K}, \epsilon)$-decision calibrated.*

This lemma implies that, to obtain an $\epsilon$-decision calibrated predictor for the function class $\mathcal{L}_\phi$, it suffices to construct an $\epsilon/2$-decision calibrated predictor for its uniform approximation class.

# 3   LOWER BOUND UNDER DETERMINISTIC OPTIMAL DECISION RULE

In this section, we investigate whether a dimension-free algorithm for decision calibration is possible under the optimal decision rule. We establish a statistical lower bound on the sample complexity of testing approximate decision calibration. In particular, we show that any algorithm requires at least $\Omega(\sqrt{d})$ samples to determine whether a predictor is decision calibrated. To the best of our knowledge, this provides the first lower bound result for decision calibration.

Note that we choose not to prove a lower bound for computing a decision-calibrated predictor directly because trivial solutions—such as a constant predictor always outputting the mean outcome $\mathbb{E}[Y]$—can satisfy both decision and full calibration.

To prove our lower bound, we consider a simple setting where the number of actions $|\mathcal{A}| = 2$, the feature mapping is $\phi(y) = y$, the class of loss functions is linear $\mathcal{L}_{\mathrm{LIN}} = \{\ell \mid \forall a, \exists r_\ell(a), \|r_\ell(a)\|_2 \leq 1, \ell(a, y) = \langle r_\ell(a), y \rangle\}$ with their corresponding optimal decision rules $\mathcal{K}_{\mathcal{L}_{\mathrm{LIN}}}$. Our result shows that distinguishing whether a predictor $p$ (and its induced loss estimator $f$) is $(\mathcal{L}_{\mathrm{LIN}}, \mathcal{K}_{\mathcal{L}_{\mathrm{LIN}}}, 0)$-decision calibrated versus not $(\mathcal{L}_{\mathrm{LIN}}, \mathcal{K}_{\mathcal{L}_{\mathrm{LIN}}}, \epsilon)$-decision calibrated requires sample complexity that depends polynomially on the dimension of $\mathcal{Y}$. Since the proof involves constructing multiple distributions, we will slightly abuse notation and add another argument for $\mathcal{D}$ in the definition of decision calibration error, that is

$$\mathrm{decCE}_{\mathcal{L}, \mathcal{K}}(f, \mathcal{D}) := \sup_{\ell \in \mathcal{L}, \, k \in \mathcal{K}} \left| \mathbb{E}_{(x,y) \sim \mathcal{D}} \mathbb{E}_{a \sim k(x)} [\ell(a, y)] - \mathbb{E}_{(x,y) \sim \mathcal{D}} \mathbb{E}_{a \sim k(x)} [f(x, a, \ell)] \right|.$$

For simplicity, we consider the special case where $\mathcal{X} = \mathcal{Y}$ and the predictor $p$ is the identity function, i.e., $p(x) = x$, and so input data take the form $(p(x_1), y_1), (p(x_2), y_2), \ldots, (p(x_n), y_n)$. Now we are ready to present the lower bound result:

**Theorem 3.1.** *Let $\epsilon \in (0, 1/3)$, $\mathcal{Y} = \{y \in \mathbb{R}^d \mid \|y\|_2 \leq 1\}$, and $f_p$ be the loss estimator induced by some predictor $p: \mathcal{X} \to \mathcal{Y}$. Let $A$ be any algorithm that takes samples $(p(x_1), y_1), (p(x_2), y_2), \ldots, (p(x_n), y_n)$ drawn i.i.d. from a distribution $\mathcal{D}$. Suppose that $A$ is guaranteed to output "accept" with probability at least $2/3$ whenever $\mathrm{decCE}_{\mathcal{L}_{\mathrm{LIN}}, \mathcal{K}_{\mathrm{LIN}}}(f, \mathcal{D}) = 0$ and guaranteed to output "reject" with probability at least $2/3$ whenever $\mathrm{decCE}_{\mathcal{L}_{\mathrm{LIN}}, \mathcal{K}_{\mathrm{LIN}}}(f_p, \mathcal{D}) \geq \epsilon$. Then the sample size $n \geq \Omega(\sqrt{d})$.*

The proof of Theorem 3.1 follows an indistinguishability argument akin to that of Gopalan et al. (2024a): given a predictor $p$, we construct two nearly identical distributions, $\mathcal{D}_1$ and $\mathcal{D}_2$, such that only $\mathcal{D}_1$ satisfies decision calibration. We show that distinguishing which of the two distributions generated the data requires at least $\Omega(\sqrt{m})$ samples. However, our setting departs significantly from Gopalan et al. (2024a), who study lower bounds for full calibration, which is stronger than decision calibration. As a result, our construction of $\mathcal{D}_1$ and $\mathcal{D}_2$ differs substantially and leverages the geometry of best response regions. When the action set $\mathcal{A}$ consists of two actions, these regions correspond to half-spaces of the form $\mathbf{1}[\langle r, p(x) \rangle > 0]$. The core idea behind constructing $\mathcal{D}_2$ is to introduce a subtle bias in the outcomes–specifically, a deviation $(y - p(x))$–that is statistically difficult to distinguish from zero-mean label noise. Simultaneously, using a "shattering argument" from VC theory, we show the existence of a loss function $\ell$ such that the associated half-space captures a biased region. Consequently, the predictor $p$ fails to satisfy decision calibration under $\mathcal{D}_2$. We leave the formal proof of Theorem 3.1 in Appendix I.

This lower bound result also exhibit a barrier result for a dimension-free algorithm for achieving decision calibration under the deterministic optimal decision rule. All existing decision calibration algorithms with provable guarantees proceed by iteratively post-processing an initial predictor $p_0$. A key component of these algorithms is the *auditing* step, which, in each iteration, identifies loss functions that witness large decision calibration error whenever the predictor is not calibrated, and returns nothing when the predictor is already calibrated (Zhao et al., 2021; Gopalan et al., 2022b; 2024a). Note that any auditing algorithm will necessarily require $\Omega(\sqrt{d})$ sample complexity based on Theorem 3.1.

## 4 AUDITING OF DECISION CALIBRATION FOR FUNCTIONS IN RKHS

In Section 3, we showed that under the optimal decision rule, it is impossible to even determine whether a predictor is decision calibrated. This provides strong evidence that it is unlikely that a non-trivial decision calibrated predictor can be learned with sample complexity independent of the dimension $m$. We now present our first positive result. By instead focusing on the smooth optimal decision rule, it becomes possible to design an algorithm with sample complexity independent of $m$. In this section, we first address the auditing problem—that is, identifying a pair of loss functions $(\ell, \ell')$ that violate decision calibration—and show that this can be achieved with sample complexity independent of $m$.

We focus on the auditing problem for functions in RKHS, since RKHS is the most general feature space considered in our paper. In detail, let $\mathcal{H}$ denote an RKHS associated with the kernel function $K : \mathcal{Y} \times \mathcal{Y} \to \mathbb{R}$. From now on, for simplicity we restrict attention to loss functions in $\mathcal{H}$ with bounded norms, that is, we define $\mathcal{L}_{\mathcal{H}} = \{\ell : \forall a, \ell(a, \cdot) \in \mathcal{H}, \|\ell(a, \cdot)\|_{\mathcal{H}} \le R_1\}$. From Lemma 2.1, the result will naturally generalize to the loss function classes that cannot be exactly represented by bounded norm functions in $\mathcal{H}$ but well approximated by them (while having an extra approximation error in the error bound). For consistency of notation, we use $r_{\ell}(a)$ to denote $\ell(a, \cdot)$. Let $\phi : \mathcal{Y} \to \mathcal{H}$ be the feature mapping induced by kernel $K$, i.e. $\phi(y) = K(y, \cdot)$ and assume that $\|\phi(y)\|_{\mathcal{H}} \le R_2$.

To ensure the loss estimator is computationally realizable, we constrain all predictors $p : \mathcal{X} \to \mathcal{H}$ to lie in the span of finitely many feature mappings,

$$p(x) = \sum_{i=1}^{N_p} \alpha_i(x) \phi(y_i), \tag{2}$$

where $N_p$ is the number of samples for estimation and $\alpha_i : \mathcal{X} \to \mathbb{R}$ is a coefficient function for any $i \in [N_p]$. For any predictor in the aforementioned form, we can define the loss estimator $f_p$ as $f_p(x, a, \ell) = \langle r_{\ell}(a), p(x) \rangle_{\mathcal{H}} = \sum_{i=1}^{N_p} \alpha_i(x) \ell(a, y_i)$.

We focus on the smoothed decision rule $\tilde{k}_{f_p, \ell}$ defined in Definition 2.4. Let $\tilde{\mathcal{K}}_{\mathcal{L}_{\mathcal{H}}}$ denote the class of such smoothed decision rules. We first show that decision calibration (Definition 2.5) for $f_p$ has an equivalent but more intuitive formulation.

**Lemma 4.1.** *For a loss estimator $f_p$ derived from the predictor $p$, it is $(\mathcal{L}_{\mathcal{H}}, \tilde{\mathcal{K}}_{\mathcal{L}_{\mathcal{H}}}, \epsilon)$-decision calibrated if and only if*

$$\sup_{\ell, \ell' \in \mathcal{L}_{\mathcal{H}}} \left| \mathbb{E}_{(x,y) \sim \mathcal{D}} \left[ \sum_{a=1}^{|\mathcal{A}|} \langle r_{\ell}(a), \phi(y) - p(x) \rangle \tilde{k}_{f_p, \ell'}(x, a) \right] \right| \le \epsilon. \tag{3}$$

An auditing algorithm is supposed to verify decision calibration with finite samples. In particular, the auditing problem asks whether we can witness a violation of decision calibration by explicitly presenting a pair of loss functions $(\ell, \ell')$ that exposes miscalibration according to Equation (3).

**Definition 4.1** (Auditing). *An $\epsilon$-auditing algorithm (or $\epsilon$-auditor) takes $(p(x_1), y_1), ...(p(x_n), y_n)$ as input, when $\mathrm{decCE}(f_p, \mathcal{D}) \ge \epsilon$, with probability $1 - \delta$, it witnesses a pair of loss functions $\ell, \ell'$, such that*

$$\left| \mathbb{E}_{(x,y) \sim \mathcal{D}} \left[ \sum_{a=1}^{|\mathcal{A}|} \langle r_{\ell}(a), \phi(y) - p(x) \rangle \tilde{k}_{f_p, \ell'}(x, a) \right] \right| \ge \epsilon/2.$$

As the first step of developing a dimension-free auditing algorithm, we establish a uniform convergence bound for the audited error of decision calibration. The upper bound is polynomial with respect to $|\mathcal{A}|$ and $1/\epsilon$, while being independent of the dimension of $\mathcal{H}$. We consider a function class parameterized by a pair of losses $(\ell, \ell')$, which collects all the possible calibration gap functions.

$$\mathcal{G} := \{g_{\ell,\ell'} : \ell, \ell' \in \mathcal{L}_\mathcal{H}\}, \quad \text{where } g_{\ell,\ell'}(p(x), \phi(y)) = \sum_{a=1}^{|\mathcal{A}|} \langle r_\ell(a), \phi(y) - p(x)\rangle \tilde{k}_{f_p,\ell'}(x,a). \quad (4)$$

We establish the uniform convergence property for $\mathcal{G}$ by showing that the covering number of $\mathcal{G}$ remains bounded, even when $\mathcal{H}$ is infinite-dimensional. Formally, we state the following theorem.

**Theorem 4.1.** *Let $D = \{(x_1, y_1), ..., (x_n, y_n)\}$ be the dataset where $(x_i, y_i)$ is drawn i.i.d. from $\mathcal{D}$. Given any predictor $p : \mathcal{X} \to \mathcal{H}$, for the function class $\mathcal{G}$ defined in Equation (4), we have that*

$$\sup_{g_{\ell,\ell'} \in \mathcal{G}} \left| \mathbb{E}_{(x,y)\sim\mathcal{D}}[g_{\ell,\ell'}(p(x), \phi(y))] - \mathbb{E}_{(x,y)\sim D}[g_{\ell,\ell'}(p(x), \phi(y))] \right| \leq O\left( \frac{C_1 \log(C_2 n) + \log(1/\delta)}{\sqrt{n}} \right),$$

*where $C_1 = |\mathcal{A}|^{\frac{3}{2}} \beta^2 R_1^3 R_2^3$ and $C_2 = R_1 R_2$.*

*Proof sketch.* By standard Rademacher argument, it suffices to prove class $\mathcal{G}$ has dimension-free finite Rademacher complexity. Since each function in the class $\mathcal{G}$ is parameterized by a pair of loss functions $\ell, \ell' \in \mathcal{L}_\mathcal{H}$, Dudley's chaining technique implies that it is enough to upper bound the covering number $N(\mathcal{L}_\mathcal{H} \times \mathcal{L}_\mathcal{H}, L_2^\mathcal{G}(P_n), \epsilon)$ where $P_n$ is the uniform distribution over dataset $D$ and $L_2^\mathcal{G}(P_n)$ is defined as

$$L_2^\mathcal{G}(P_n)\left((\ell^1, \ell^{1'}), (\ell^2, \ell^{2'})\right) := \sqrt{\frac{1}{n} \sum_{i=1}^n \left(g_{\ell^1,\ell^{1'}}(p(x_i), \phi(y_i)) - g_{\ell^2,\ell^{2'}}(p(x_i), \phi(y_i))\right)^2}.$$

In order to bound the covering number $N(\mathcal{L}_\mathcal{H} \times \mathcal{L}_\mathcal{H}, L_2^\mathcal{G}(P_n), \epsilon)$, observe that for any $\ell \in \mathcal{L}_\mathcal{H}$ and any $a \in \mathcal{A}$, $r_\ell(a)$ is in the Hilbert ball $B(R_1)$ with radius $R_1$ since $\|r_\ell(a)\|_\mathcal{H} \leq R_1$. This allows us to connect the covering number we want to bound to a *known finite* covering number $N(B(R_1), d_P, \epsilon)$ where $P$ is an arbitrary distribution on the Hilbert ball and $d_P$ is defined as

$$d(r_{\ell_1}(a), r_{\ell_2}(a)) = \sqrt{\mathbb{E}_{X\sim P}[\langle r_{\ell_1}(a) - r_{\ell_2}(a), X\rangle^2]},$$

where $X$ is a random sample in the Hilbert ball drawn from distribution $P$. Intuitively, $d_P$ first projects the differences $\theta - \theta'$ along a random direction given by the prediction of a random example $p(X)$ and then measures the distances in this projected one-dimensional space. For the formal proof, see Appendix K. $\square$

As a corollary of Theorem 4.1, we can develop an ERM oracle that serves as an auditing algorithm. The loss function of the ERM is defined as

$$L_{\text{DecCal}}(\ell, \ell', x, y) = g_{\ell,\ell'}(p(x), \phi(y)) = \sum_{a=1}^{|\mathcal{A}|} \langle r_\ell(a), \phi(y) - p(x)\rangle \tilde{k}_{f_p,\ell'}(x,a).$$

**Theorem 4.2** (ERM as Auditing Algorithm). *Let $D = \{(x_1, y_1), ..., (x_n, y_n)\}$ be the dataset that each data point is drawn i.i.d. from $\mathcal{D}$, given any predictor $p : \mathcal{X} \to \mathcal{H}$, the ERM algorithm that outputs*

$$\hat{\ell}, \hat{\ell}' \leftarrow \arg\max_{\ell,\ell'} \frac{1}{n} \sum_{i=1}^n L_{\text{DecCal}}(\ell, \ell', x_i, y_i),$$

*when $n \geq \tilde{O}(|\mathcal{A}|^3 \beta^4 R_1^6 R_2^6 \epsilon^{-2})$, ERM algorithm is an $\epsilon$-auditor.*

In fact, solving ERM is stronger than solving the auditing problem. Auditing does not require identifying the pair of loss functions that maximizes the empirical decision calibration error; it suffices to find a pair for which the empirical error is large enough to certify that the true expected decision calibration error exceeds $\epsilon/2$ (Definition 4.1). To avoid potential misinterpretation, our algorithm Algorithm 1 assumes only the existence of an auditing oracle, rather than requiring an ERM oracle.

## 5 ALGORITHMS OF DECISION CALIBRATION FOR FUNCTIONS IN RKHS

In this section, we discuss algorithms of decision calibration for functions in RKHS. In Section 5.1, we first present our algorithm `DimFreeDeCal` (Algorithm 1) to achieve $(\mathcal{L}_{\mathcal{H}}, \tilde{\mathcal{K}}_{\mathcal{L}_{\mathcal{H}}}, \epsilon)$-decision calibration. The *patching* component of our algorithm is motivated by the weighted calibration framework introduced by Gopalan et al. (2022b), based on the observation that decision calibration can be viewed as a special case of weighted calibration. However, their algorithmic framework is not directly applicable to our setting, as it patches the predictor in the finite-dimensional setting, whereas our formulation requires handling a more general (potentially infinite-dimensional) prediction space. We will discuss how to address challenges in the infinite-dimensional setting.

Zhao et al. (2021) also proposed an algorithm for achieving decision calibration under the smooth optimal decision rule in the finite-dimensional setting, given the access to the full data distribution. However, their algorithm does not directly extend to the finite-sample or infinite-dimensional settings. In Appendix M, we describe how to adapt their algorithm to achieve provable finite-sample guarantees and extend it to the infinite-dimensional case. Notably, our proposed algorithm achieves a sample complexity of $\tilde{O}(1/\epsilon^4)$, which improves upon the $\tilde{O}(1/\epsilon^6)$ sample complexity of the modified version of their algorithm.

### 5.1 DIMENSION-FREE DECISION CALIBRATION ALGORITHM

In this section, we propose our algorithm `DimFreeDeCal` (Algorithm 1). In Section 5.1.1, we build the connection between decision calibration and weighted calibration. Building on this connection, we address the novel challenges of patching in the infinite-dimensional setting and present our algorithm in Section 5.1.2.

#### 5.1.1 DECISION CALIBRATION AS WEIGHTED CALIBRATION

We restate the definition of weighted calibration introduced by Gopalan et al. (2022b) and extend it to the RKHS setting.

**Definition 5.1** (Weighted Calibration Gopalan et al. (2022b))**.** *Let $\mathcal{W} : \mathcal{H} \to \mathcal{H}$ be a family of weight functions. We define the $\mathcal{W}$-calibration error as*

$$\mathrm{CE}_{\mathcal{W}}(p) = \sup_{w \in \mathcal{W}} |\mathbb{E}_{\mathcal{D}}[\langle w(p(x)), p(x) - \phi(y)\rangle_{\mathcal{H}}]|.$$

*We say that the loss estimator $f_p$ is $(\mathcal{W}, \epsilon)$-calibrated if $\mathrm{CE}_{\mathcal{W}}(p) \leq \epsilon$.*

The weighted calibration algorithm follows an iterative template: at round $t$,

1. Use an auditing algorithm to check whether $p_t$ is $(\mathcal{W}, \epsilon)$-weighted calibrated. If it is, terminate the algorithm.
2. If not, identify the weight function $w_t \in \mathcal{W}$ that incurs the largest $\mathcal{W}$-calibration error.
3. Update the predictor $p_{t+1}(x) := p_t(x) + \eta \cdot w_t(p_t(x))$, where $\eta$ is the step-size hyperparameter.

Next we will show the connection between decision calibration and weighted calibration. By Lemma 4.1, the decision calibration error of a loss estimator $f_p$ can be written as

$$\mathrm{decCE}_{\mathcal{L}_{\mathcal{H}}, \tilde{\mathcal{K}}_{\mathcal{L}_{\mathcal{H}}}}(f_p) := \sup_{\ell \in \mathcal{L}_{\mathcal{H}}, \tilde{k} \in \tilde{\mathcal{K}}_{\mathcal{L}_{\mathcal{H}}}} \left| \mathbb{E}_{(x,y)\sim\mathcal{D}}\mathbb{E}_{a\sim\tilde{k}(x)}[\ell(a,y)] - \mathbb{E}_{(x,y)\sim\mathcal{D}}\mathbb{E}_{a\sim\tilde{k}(x)}[f(x,a,\ell)] \right|$$

$$= \sup_{\ell,\ell' \in \mathcal{L}_{\mathcal{H}}} \left| \mathbb{E}_{(x,y)\sim\mathcal{D}} \left[ \sum_{a=1}^{|\mathcal{A}|} \langle r_\ell(a), \phi(y) - p(x)\rangle \tilde{k}_{f_p,\ell'}(x,a) \right] \right|$$

$$= \sup_{\ell,\ell' \in \mathcal{L}_{\mathcal{H}}} \left| \mathbb{E}_{(x,y)\sim\mathcal{D}} \left[ \left\langle \sum_{a=1}^{|\mathcal{A}|} r_\ell(a)\tilde{k}_{f_p,\ell'}(x,a), \phi(y) - p(x) \right\rangle \right] \right|.$$

Therefore, decision calibration is a special instance of $\mathcal{W}_{\mathrm{dec}}$-calibration for $\mathcal{W}_{\mathrm{dec}} := \{w_{\ell,\ell'} : w_{\ell,\ell'}(p(x)) = \sum_{a=1}^{|\mathcal{A}|} r_\ell(a)\tilde{k}_{f_p,\ell'}(x,a), \forall \ell, \ell' \in \mathcal{L}_{\mathcal{H}}\}$.

### 5.1.2 PATCHING IN THE INFINITE-DIMENSIONAL SETTING

The first challenge in the infinite-dimensional setting is that we need to restrict the predictor in the form of Eq. (2) so that we can use the reproducing property to construct a loss estimator $f_p$. Therefore, in each round $t$, once we find $\ell_t, \ell_t'$ that violates the decision calibration, we cannot directly follow the original weighted calibration algorithm template to update the predictor by patching $w_{\ell_t, \ell_t'}$ *unless* $r_{\ell_t}(a)$ can be explicitly expressed by the linear combination of $\phi(y)$.

However, note that

$$\left|\mathbb{E}\left[\left\langle \sum_{a=1}^{|\mathcal{A}|} r_{\ell_t}(a)\tilde{k}_{\ell_t'}(x,a), \phi(y) - p_t(x)\right\rangle\right]\right| = \left|\sum_{a=1}^{|\mathcal{A}|}\left\langle r_{\ell_t}(a), \mathbb{E}[(\phi(y) - p_t(x))\tilde{k}_{\ell_t'}(x,a)]\right\rangle\right|$$

$$\leq \sum_{a=1}^{|\mathcal{A}|} R_1\left\|\mathbb{E}[(\phi(y) - p_t(x))\tilde{k}_{\ell_t'}(x,a)]\right\|_{\mathcal{H}}.$$

Equality holds when $r_{\ell_t^*}(a) = R_1\mathbb{E}[(\phi(y) - p_t(x))\tilde{k}_{\ell_t'}(x,a)]/\|\mathbb{E}[(\phi(y) - p_t(x))\tilde{k}_{\ell_t'}(x,a)]\|_{\mathcal{H}}$. On the one hand, once we identify a pair of $(\ell_t, \ell_t')$, replacing $\ell_t$ with $\ell_t^*$ will make the violation worse. On the other hand, $r_{\ell_t^*}(a)$ can be expressed by the linear combination of $\phi(y)$ (we will use the empirical expectation to approximate the true expectation). Therefore, in each round $t$, we can use $\ell_t^*$ to update the predictor $p_t$.

---

**Algorithm 1** DimFreeDeCal

**Input:** The RKHS kernel $K$, current predictor $p_0 : \mathcal{X} \to \mathcal{H}$, tolerance $\epsilon$ and step $t = 0$.

1: **while** $\sup_{\ell, \ell'} \mathbb{E}[\langle \sum_{a=1}^{|\mathcal{A}|} r_\ell(a)\tilde{k}_{\ell'}(x,a), \phi(y) - p_t(x)\rangle] > \epsilon$ **do**

2:     Find $\ell_t, \ell_t'$ such that $\hat{\mathbb{E}}[\langle \sum_{a=1}^{|\mathcal{A}|} r_{\ell_t}(a)\tilde{k}_{\ell_t'}(x,a), \phi(y) - p_t(x)\rangle] > 3\epsilon/4$.

3:     Define the adjustments

$$d_{ta} = \eta R_1\hat{\mathbb{E}}[(\phi(y) - p_t(x))\tilde{k}_{\ell_t'}(x,a)]/\left\|\hat{\mathbb{E}}[(\phi(y) - p_t(x))\tilde{k}_{\ell_t'}(x,a)]\right\|_{\mathcal{H}}.$$

4:     Set $p_{t+1} : x \mapsto p_t(x) + \sum_{a=1}^{|\mathcal{A}|} d_{ta}\tilde{k}_{\ell_t'}(x,a)$.

5:     Set $p_{t+1} : x \mapsto \pi_{B(R_2)}(p_{t+1}(x))$.     $//\pi_{B(R_2)}$ *projects onto Hilbert ball* $B(R_2)$.

6: **end while**

---

Intuitively, the algorithm proceeds as follows. In lines 2–3, we invoke the *auditing* oracle: if the loss estimator $f_{p_0}$ is not $(\mathcal{L}_{\mathcal{H}}, \tilde{K}_{\mathcal{L}_{\mathcal{H}}}, \epsilon)$-decision calibrated, we can identify a pair $(\ell_t, \ell_t')$ that the empirical decision calibration error exceeds $3\epsilon/4$. In line 4, as previously discussed, we substitute $(\ell_t^*, \ell_t')$ for $(\ell_t, \ell_t')$ to define the patching term so that the updated predictor can remain to be explicitly expressed as a linear combination of $\phi(y)$. Lines 5–6 then carry out the patching step. Notably, we cannot perform computations directly with $p_t(x)$, as it may reside in an infinite-dimensional space. To address this, we introduce the technique of *implicit patching*. The key idea is to perform patching implicitly by maintaining a linear representation of the form $p_t(x) = \sum_{i=1}^{N_t} \alpha_{ti}(x)\phi(y_{ti})$. That is, we keep track of the functions $\alpha_{ti} : \mathcal{X} \to \mathbb{R}$ and the corresponding outcomes $y_{ti}$ for all $t$ and $i \in [N_t]$. Given this representation, we can efficiently compute the value of the loss estimator $f_{p_t}(x, a, \ell)$ for any loss function $\ell \in \mathcal{L}_{\mathcal{H}}$ as follows $f_{p_t}(x, a, \ell) = \langle r_\ell(a), p_t(x)\rangle_{\mathcal{H}} = \sum_{i=1}^{N_t} \alpha_{ti}(x)\ell(a, y_{ti})$. Formally, we have the following proposition.

**Proposition 5.1.** *For Algorithm 1, if the input predictor satisfies $p_0(x) = \sum_{i=1}^{N_0} \alpha_{0i}(x)\phi(y_{0i})$, the for any $t$, we have $p_t(x) = \sum_{i=1}^{N_t} \alpha_{ti}(x)\phi(y_{ti})$.*

Given Proposition 5.1, we can perform patching implicitly by keeping track of the coefficients instead of directly computing $p_t$.

Now we are ready to present the main result of this section.

**Theorem 5.1.** *Given any initial predictor $p_0$ and tolerance $\epsilon$, Algorithm 1 terminates in $T = O(\frac{R_1^2 R_2^2}{\epsilon^2})$ iterations. Given $\tilde{O}(\frac{|\mathcal{A}|^3 R_1^8 R_2^8}{\epsilon^4})$ samples, with probability $1 - \delta$, Algorithm 1 outputs*

*a predictor $p_T$ such that $f_{p_T}$ is $(\mathcal{L}_{\mathcal{H}}, \tilde{\mathcal{K}}_{\mathcal{L}_{\mathcal{H}}}, \epsilon)$-decision calibrated and* $\mathbb{E}[\|p_T(x) - \phi(y)\|_{\mathcal{H}}^2] \leq \mathbb{E}[\|p_0(x) - \phi(y)\|_{\mathcal{H}}^2]$.

This theorem establishes that Algorithm 1 produces a decision-calibrated predictor that preserves the performance of the initial predictor, requiring only finitely many samples that do not depend on the dimension $m$.

To conclude, this paper investigates when model predictions can be made *decision-calibrated* for nonlinear downstream losses, especially when naive embeddings $y \mapsto \phi(y) \in \mathbb{R}^m$ make $m$ large or infinite. We first show a limitation: under deterministic best responses, even *verifying* decision calibration can require sample complexity polynomial in $m$. We then introduce a smooth variant and an accompanying algorithm that achieves decision calibration with $\text{poly}(|\mathcal{A}|, 1/\varepsilon)$ samples independent of $m$, without degrading the base predictor in terms of $\ell_2$ losses, and covers broad nonlinear losses via bounded-norm functions in separable RKHS.

## ACKNOWLEDGEMENTS

ZSW was partially supported by an NSF CAREER Award, an NSF SaTC grant, and an STTR grant. We thank Aaron Roth for feedback on an earlier draft of this work and Lunjia Hu for helpful suggestions on the lower bound.

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

## A   ETHICS STATEMENT

This work is entirely theoretical and does not involve human subjects, sensitive personal data, or potentially harmful applications. As such, we do not foresee any direct ethical concerns.

## B   REPRODUCIBILITY STATEMENT

We have provided complete formal definitions, theorems, and proofs in the main text and appendix to ensure the reproducibility of our results. All assumptions are explicitly stated, and key lemmas and lower bound constructions are detailed. Since our contributions are theoretical, no datasets are involved, and all algorithmic procedures are described in a way that can be unambiguously implemented by others.

## C   USE OF LARGE LANGUAGE MODELS

We used LLMs solely as a writing assistant to polish the presentation and improve readability.

## D   USEFUL LEMMAS

**Lemma D.1** (Property of Traces and Frobenius norms). *For any matrix $A \in \mathbb{R}^{m \times n}$, the Frobenius norm is defined as*

$$\|A\|_F = \sqrt{\sum_{i=1}^{m} \sum_{j=1}^{n} a_{ij}^2}.$$

*We have*

1. $\mathrm{Tr}(AA^T) = \mathrm{Tr}(A^T A) = \|A\|_F^2$.

2. *When $A, B$ are square matrices,* $\mathrm{Tr}(AB) \leq \sqrt{\mathrm{Tr}(AA^T) \cdot \mathrm{Tr}(BB^T)} = \|A\|_F \|B\|_F$.

3. *Frobenius norm has submultiplicative property, that is, for any matrix $A, B$,*

$$\|AB\|_F \leq \|A\|_F \|B\|_F.$$

**Theorem D.1** (Hoeffding's Inequality for Hilbert Spaces). *Let $\mathcal{H}$ be a separable Hilbert space. Let $X_1, \cdots, X_N$ be independent random elements of $\mathcal{H}$ with common mean $\mu$ such that $\|X_i\| \leq B$ almost surely for any $i \in [N]$. Let $\hat{\mu}_N := \frac{1}{N} \sum_{i=1}^N X_i$ denote the sample mean. Then for any $\delta \in (0,1)$ with probability at least $1 - \delta$,*

$$\|\hat{\mu}_N - \mu\| \leq 2B \sqrt{\frac{2\ln(2/\delta)}{N}}.$$

We will introduce some useful results for proving the uniform convergence guarantee.

**Definition D.1** (Covering Numbers). *Let $(V, d)$ be a metric space and $\Theta \subset V$. We say $\{v_i\}_{i=1}^N \subset V$ is an $\epsilon$-covering of $\Theta$ if $\Theta \subset \bigcup_{i=1}^N B(v_i, \epsilon)$ where $B(v, \epsilon) := \{u \in V : d(u, v) \leq \epsilon\}$ is the closed ball of radius $\epsilon$ centered at $v$. The covering number is defined as*

$$N(\Theta, d, \epsilon) := \min\{n : \exists \epsilon\text{-covering of } \Theta \text{ of size } n\}$$

**Definition D.2** (Rademacher Complexity). *Let $S = \{z_1, ..., z_n\} \subset Z$ be a sample of points, and a function class $\mathcal{F}$ of real-valued functions over $Z$. The Rademacher complexity of $\mathcal{F}$ with respect to $S$ is defined as follows:*

$$\mathcal{R}_S(\mathcal{F}) = \frac{1}{n} \mathbb{E}_{\sigma \sim \{-1, +1\}^m} \left[ \sup_{f \in \mathcal{F}} \sum_{i=1}^n \sigma_i f(z_i) \right]$$

**Theorem D.2.** *Assume that $z_1, ..., z_m$ are i.i.d. drawn from $\mathcal{D}$, then with probability at least $1 - \delta$, we have*

$$\sup_{f \in \mathcal{F}} \left[ \frac{1}{n} \sum_{i=1}^n f(z_i) - \mathbb{E}_{z \sim \mathcal{D}}[f(z)] \right] \leq 2\mathbb{E}_{S \sim \mathcal{D}^m}[\mathcal{R}_S(\mathcal{F})] + \sqrt{\frac{\log(2/\delta)}{2n}}$$

We consider a Hilbert ball $B_2 = \{x \in \mathbb{R}^\infty | \sum_t x_i^2 \leq 1\}$ Now we introduce the result that upper bounds the covering number of Hilbert balls under some metric induced by a probability distribution $P$. Note that under the common metric $\ell_2(\mathbb{R}^\infty)$, the covering number of the Hilbert balls is infinite. However, under the metric $d_p(\theta, \theta') = \sqrt{\mathbb{E}_{X \sim P} |\langle \theta - \theta', X \rangle|^2}$, the covering number is finite even in the infinite dimensional Hilbert space.

**Theorem D.3** (Covering Number of Hilbert Balls MacKay (2003)). *$P$ is a distribution on $B_2$, consider the metric $d_P(\theta, \theta') = \sqrt{\mathbb{E}_{X \sim P} |\langle \theta - \theta', X \rangle|^2}$. There exists a universal constant $c$, such that for any $P$, $\epsilon > 0$, we have*

$$\log N(B_2, d_P, \epsilon) \leq \frac{c}{\epsilon^2}.$$

Let $P_n$ be the empirical distribution, which is the uniform distribution over $z_1, ..., z_n$. For a function class $\mathcal{F}$, we define the metric $L_2^\mathcal{F}(P_n)(f, f') = \sqrt{\frac{1}{n} \sum_{i=1}^n (f(z_i) - f'(z_i))^2}$. Note that if you plug in $P = P_n$ for the metric $d_P$ in Theorem D.3, then the metric $d_P$ becomes a special case of $L_2^\mathcal{F}(P_n)$ for $f(z) = \langle \theta, z \rangle$.

Now we indroduce the Dudley's Theorem which bounds the Rademacher complexity of a function class by its covering number.

**Theorem D.4** (Localized Dudley's Theorem). *Let $S = \{z_1, ..., z_n\} \subset Z$ be a sample of points, and a function class $\mathcal{F}$ of real-valued functions over $Z$. For any $\alpha \geq 0$, we have*

$$\mathcal{R}_S(\mathcal{F}) \leq 4\alpha + 12 \int_\alpha^\infty \sqrt{\frac{\log N(\mathcal{F}, L_2^\mathcal{F}(P_n), \epsilon)}{n}} d\epsilon. \tag{5}$$

# E  RELATED WORK

**Calibration and Decision Making**  The work most closely related to ours is Zhao et al. (2021), which introduced the concept of *decision calibration* in the *batch setting*, where data points are drawn from an underlying distribution. Zhao et al. (2021) primarily examined decision calibration in the context of multi-class classification, where the outcome space $\mathcal{Y}$ is finite and the loss functions are linear. Our paper also considers the batch setting, but we significantly extend their framework to a broader and more general scenario, allowing the outcome space $\mathcal{Y}$ to be any compact convex set and accommodating non-linear loss functions. There is a longstanding line of work on calibration and decision-making in the *adversarial setting*, where data are presented adversarially in a sequential manner. The seminal work of Foster & Vohra (1999) showed that a decision maker who best responds to calibrated forecasts obtains diminishing internal regret. Similarly to decision calibration, there is a line of work in the adversarial setting that tries to achieve some weaker variants of calibration while keeping agents incentivized to treat the predictions as correct (Kleinberg et al. (2023); Fishelson et al. (2025); Luo et al. (2025)). Kleinberg et al. (2023) proposed a notion called *U-calibration*, which is sufficient for agents to achieve sublinear *external* regret, bypassing the lower bound of achieving calibration (Qiao & Valiant (2021)). A subsequent work by Luo et al. (2024) gave the optimal bound of multiclass U-calibration. Noarov et al. (2023) studied how to make sequential predictions for decision-making in the high-dimensional setting, but also relied on the loss functions to be linear. Following the same algorithmic approach as Noarov et al. (2023), Roth & Shi (2024) showed how to produce predictions for agents to best respond and achieve low *swap* regret. But their regret bound has dependence on the size of the action $|\mathcal{A}|$. Hu & Wu (2024) showed that in the binary setting, the dependence on $|\mathcal{A}|$ can be removed while keeping the $\tilde{O}(\sqrt{T})$ regret. There is also work on calibration and decision making in games, such as Camara et al. (2020); Haghtalab et al. (2023); Collina et al. (2024). However, most of these works focus either on linear loss functions or on one-dimensional outcome spaces, whereas our work addresses the more general and challenging setting of nonlinear loss functions over $d$-dimensional outcomes.

**Omniprediction**  In addition to decision calibration, there is another line of work studying prediction and downstream decision making called *omniprediction*, which was introduced by Gopalan et al. (2021). A subsequent Gopalan et al. (2022a) built the connection between omniprediction and outcome indistinguishability (OI), which was introduced by Dwork et al. (2021) in the binary setting and was extended to the continuous one-dimensional setting by Dwork et al. (2022). In detail, they showed that omniprediction can be achieved by *hypothesis* OI and *decision OI*. Decision OI is a weaker notion than decision calibration. While decision OI requires that predictions be indistinguishable from the true outcomes with respect to the loss $\ell$ incurred under the optimal decision rule defined by $\ell$ itself, decision calibration demands this indistinguishability hold for the loss $\ell$ incurred under the optimal decision rules defined by any loss function $\ell'$.

Garg et al. (2024) first studied omniprediction in the adversarial setting. Recently, several papers on omniprediction have leveraged decision OI to achieve omniprediction efficiently in both batch and adversarial settings. Okoroafor et al. (2025) studied near-optimal omniprediction in the adversarial binary setting. Gopalan et al. (2024b) studied how to efficiently achieve omniprediction for nonlinear losses in the one-dimensional batch setting. They proposed the notion called *sufficient statistics*, which can be viewed as finite-dimensional feature mapping and inspired our study on more general feature mapping. Dwork et al. (2024) studied omniprediction in evolving graphs. A very recent work Lu et al. (2025) extended Gopalan et al. (2024b) to the high-dimensional adversarial setting by using a different generalization of the decision OI, first given by Noarov et al. (2023) and used by Roth & Shi (2024). It is worth noting that their work is not directly comparable to ours, even though they also consider $d$-dimensional nonlinear losses, for the following reasons. First, similar to Gopalan et al. (2024b), their framework to handle nonlinear loss functions assumes a finite-dimensional feature mapping, whereas we also address the more general case of infinite-dimensional feature mappings. Second, their focus lies in the adversarial setting, where they employ an online-to-batch conversion to construct a randomized predictor from scratch that satisfies batch omniprediction. In contrast, our goal is to take an arbitrary predictor as input and output a deterministic predictor that satisfies decision calibration—a related but fundamentally different notion from omniprediction. Uniform approximation via finite-dimensional feature mappings has been studied in prior work, such as Gopalan et al. (2024b) and Lu et al. (2025), in the contexts of omnipredic-

tion and online decision swap regret. Our formulation generalizes this idea to infinite-dimensional feature spaces.

**Calibration and Reproducing Kernel Hilbert Spaces (RKHS)**   RKHS has also been introduced in the context of weighted calibration (Gopalan et al., 2022b). Błasiok et al. (2023) and Gopalan et al. (2024a) consider weight functions that are bounded-norm functions in an RKHS. However, the role of RKHS in our work is fundamentally different. While these prior works use RKHS for the weight functions, we use it to represent the loss functions. Specifically, their approach involves computing the inner product between the weight function and the difference $p(x) - y$, which preserves the original outcome space. As a result, achieving weighted calibration with bounded-norm weights in an RKHS does not translate to decision calibration for non-linear losses.

## F   LINEAR FUNCTIONS IN RKHS

**Example F.1** (Continuous Piecewise Linear Functions). *Consider the case $\mathcal{Y} = [0, 1]$. Define a family of functions to be $\mathcal{G} = \{g_{k_1, k_2, c} : \forall c \in [0, 1], |k_1| \leq R, |k_2| \leq R\}$ where*

$$g_{k_1, k_2, c}(y) = \begin{cases} k_1 y & 0 \leq y < c \\ k_2 y + (k_1 - k_2)c & c \leq y \leq 1. \end{cases}$$

*This defines a class of piecewise linear functions with an unknown turning point $c$. Piecewise linear functions of this form have been extensively studied in the economics literature. The function $g_{k_1, k_2, c}$ can be interpreted as a utility function (or the negative of a loss function), where $y$ denotes the consumption level of a particular good. It captures a common economic scenario in which marginal utility decreases once consumption exceeds a threshold $c$.*

*Next we show that functions in $\mathcal{G}$ are linear in a infinite-dimensional feature space. Let $\mathcal{H}$ be the RKHS with kernel*

$$K(y_1, y_2) = \min\{y_1, y_2\}.$$

*Let $\phi(y) := K(y, \cdot)$ be the feature mapping associated with $K$. We have*

$$g_{k_1, k_2, c} = \langle k_2 \phi(1) + (k_1 - k_2)\phi(c), \phi(y) \rangle_{\mathcal{H}}.$$

*In addition, we have $\|k_2 \phi(1) + (k_1 - k_2)\phi(c)\|_{\mathcal{H}} \leq R$.*

**Example F.2** (Cobb-Douglas Functions). *Consider the case $\mathcal{Y} = \{y \in \mathbb{R}^d : \|y\|_2 \leq 1\}$. Define a family of functions to be $\mathcal{G} = \{g_\alpha : \forall \alpha \in [0, 1]^d \text{ s.t. } \sum_{i \in [d]} a_i = 1\}$ where*

$$g_\alpha(y) = e^{\sum_{i \in [d]} \alpha_i y_i}.$$

*This defines the class of Cobb-Douglas functions in exponential form. Cobb-Douglas functions are widely used in economics. One can interpret $g_\alpha$ as a utility function (or the negative of a loss function), where $y_i$ represents the consumption level of the $i$-th good and $a_i$ is the normalized preference (see Varian & Varian (1992)) for the $i$-th good for any $i \in [d]$.*

*Next, we show that functions in $\mathcal{G}$ are linear in an infinite-dimensional feature space. Let $\mathcal{H}$ be the RKHS with kernel*

$$K(y_1, y_2) = \exp(\langle y_1, y_2 \rangle).$$

*Let $\phi(y) := K(y, \cdot)$ be the feature mapping associated with $K$. We have*

$$g_\alpha = \langle \phi(a), \phi(y) \rangle_{\mathcal{H}}.$$

*In addition, we have $\|\phi(a)\|_{\mathcal{H}} \leq \sqrt{e}$.*

## G   NO REGRET GUARANTEES OF DECISION CALIBRATION

Now we show why decision calibration is useful, as it gives no regret guarantees for downstream decision makers. We consider the no-type-regret guarantee that is also discussed in Zhao et al. (2021). Informally, no type-regret guarantee ensures that a decision maker with loss function $\ell \in \mathcal{L}$, who plays the best response policy under their own loss, will incur an expected loss no greater than

what they would incur by playing the best response strategy for any other loss function $\ell' \in \mathcal{L}$.[1] We derive the no-type-regret guarantee results for decision makers under both the optimal decision rule and the smooth optimal decision rule.

**Proposition G.1** (No Type Regret under Optimal Decision Rule). *If the loss estimator $f_p$ is $(\mathcal{L}, \mathcal{K}_{\mathcal{L}}, \epsilon)$-decision calibrated, then any decision maker under optimal decision rule has no regret reporting their true loss function, that is*

$$\forall \ell, \ell' \in \mathcal{L}, \mathbb{E}_{(x,y) \sim \mathcal{D}} \mathbb{E}_{a \sim k_\ell(x)}[\ell(a,y)] \leq \mathbb{E}_{(x,y) \sim \mathcal{D}} \mathbb{E}_{a \sim k_{\ell'}(x)}[\ell(a,y)] + 2\epsilon.$$

*Proof.* By definition, when the loss estimator $f_p$ is $(\mathcal{L}, \mathcal{K}_{\mathcal{L}}, \epsilon)$-decision calibrated, we have

$$\mathbb{E}_{(x,y) \sim \mathcal{D}} \mathbb{E}_{a \sim k_\ell(x)}[\ell(a,y)]$$
$$\leq \mathbb{E}_{(x,y) \sim \mathcal{D}} \mathbb{E}_{a \sim k_\ell(x)}[f(x,a,\ell)] + \epsilon$$
$$\leq \mathbb{E}_{(x,y) \sim \mathcal{D}} \mathbb{E}_{a \sim k_{\ell'}(x)}[f(x,a,\ell)] + \epsilon$$
$$\leq \mathbb{E}_{(x,y) \sim \mathcal{D}} \mathbb{E}_{a \sim k_{\ell'}(x)}[\ell(a,y)] + 2\epsilon,$$

where the first and third inequalities follow from the definition of decision calibration, and the second inequality follows from the optimality of $k_\ell$. □

Zhao et al. (2021) proved a similar guarantee for multiclass setting and linear loss function class. We generalize the result to the general loss estimator setting.

Now we move on to a similar guarantee for decision makers with the smooth optimal decision rule. For this result the error will have another term $\frac{\log(|A|)+1}{\beta}$ which is related to the hyperparameter $\beta$ of inverse-temperature. This is because the smooth best response rule $\tilde{k}_\ell$ might not strictly lead to a better expected loss than $\tilde{k}'_\ell$, therefore we will need to first relate the loss that the decision maker incurs by playing $\tilde{k}_\ell$ to the loss they incurs by playing the strict optimal decision rule $k_\ell$, which adds another approximation error term. To prove the result, we will need to use a lemma proposed by Roth & Shi (2024), where they studied swap regret (a different notion of regret) in the adversarial online setting. Roth & Shi (2024) states the lemma in the setting of a utility function $u$, and we restate it in the form of the loss function $\ell$.

**Lemma G.1** (Roth & Shi (2024)). *For any loss estimator $f$, context $x$ and loss function $\ell$, we have that*

$$\mathbb{E}_{a \sim \tilde{k}_\ell(x)}[f(x,a,\ell)] \leq \mathbb{E}_{a \sim k_\ell(x)}[f(x,a,\ell)] + \frac{\log(|A|)+1}{\beta}.$$

**Proposition G.2** (No Type Regret under Smooth Optimal Decision Rule). *If the loss estimator $f_p$ is $(\mathcal{L}, \tilde{\mathcal{K}}_{\mathcal{L}}, \epsilon)$-decision calibrated, then any decision maker under smooth optimal decision rule has no regret reporting their true loss function, that is*

$$\forall \ell, \ell' \in \mathcal{L}, \mathbb{E}_{(x,y) \sim \mathcal{D}} \mathbb{E}_{a \sim \tilde{k}_\ell(x)}[\ell(a,y)] \leq \mathbb{E}_{(x,y) \sim \mathcal{D}} \mathbb{E}_{a \sim \tilde{k}_{\ell'}(x)}[\ell(a,y)] + 2\epsilon + \frac{\log(|A|)+1}{\beta}.$$

*Proof.*

$$\mathbb{E}_{(x,y) \sim \mathcal{D}} \mathbb{E}_{a \sim \tilde{k}_\ell(x)}[\ell(a,y)]$$
$$\leq \mathbb{E}_{(x,y) \sim \mathcal{D}} \mathbb{E}_{a \sim \tilde{k}_\ell(x)}[f(x,a,\ell)] + \epsilon$$
$$\leq \mathbb{E}_{(x,y) \sim \mathcal{D}} \mathbb{E}_{a \sim k_\ell(x)}[f(x,a,\ell)] + \epsilon + \frac{\log(|A|)+1}{\beta}$$
$$\leq \mathbb{E}_{(x,y) \sim \mathcal{D}} \mathbb{E}_{a \sim k_{\ell'}(x)}[f(x,a,\ell)] + \epsilon + \frac{\log(|A|)+1}{\beta}$$
$$\leq \mathbb{E}_{(x,y) \sim \mathcal{D}} \mathbb{E}_{a \sim \tilde{k}_{\ell'}(x)}[f(x,a,\ell)] + \epsilon + \frac{\log(|A|)+1}{\beta}$$

---

[1]From the perspective of mechanism design, no-type-regret implies that decision makers have no incentives to misreport their loss function to the loss estimator.

$$\leq \mathbb{E}_{(x,y)\sim\mathcal{D}}\mathbb{E}_{a\sim\tilde{k}_{\ell'}(x)}[\ell(a,y)] + 2\epsilon + \frac{\log(|A|)+1}{\beta},$$

where the first and last inequalities follows from the definition of decision calibration, the second inequality follows from Lemma G.1, the third inequality follows from the optimality of $k_\ell$, and the fourth inequality follows from the expected loss of playing the optimal decision rule will lead to loss no greater than that when playing the smooth optimal decision rule. $\square$

# H  PROOFS IN SECTION 2

**Lemma 2.1.** *Let $\mathcal{L}_\phi$ denote the class of loss functions for which the feature mapping $\phi : \mathcal{Y} \to \mathcal{H}$ gives $(\dim(\mathcal{H}), \lambda, \frac{\epsilon}{2})$-uniform approximations and let $\hat{\mathcal{L}}_\phi = \{\hat{\ell} : \hat{\ell}(a,y) = r_\ell(a) \cdot \phi(y)\}$ denote the associated class of linear functions. For any predictor $p : \mathcal{X} \to \mathcal{H}$, any class of decision rule $\mathcal{K}$ and $\epsilon > 0$, if the loss estimator $f_p$ is $(\hat{\mathcal{L}}_\phi, \mathcal{K}, \epsilon/2)$-decision calibrated, then $f_p$ is $(\mathcal{L}_\phi, \mathcal{K}, \epsilon)$-decision calibrated.*

*Proof.* We have for any $\ell \in \mathcal{L}_\phi$ and any $k \in \mathcal{K}$,

$$\begin{aligned}
&\left|\mathbb{E}_{(x,y)\sim\mathcal{D}}\mathbb{E}_{a\sim k(x)}[\ell(a,y)] - \mathbb{E}_{(x,y)\sim\mathcal{D}}\mathbb{E}_{a\sim k(x)}[f_p(x,a,\ell)]\right| \\
&= \left|\mathbb{E}_{(x,y)\sim\mathcal{D}}\mathbb{E}_{a\sim k(x)}[\ell(a,y) - \hat{\ell}(a,y)] + \mathbb{E}_{(x,y)\sim\mathcal{D}}\mathbb{E}_{a\sim k(x)}[\hat{\ell}(a,y) - f_p(x,a,\ell)]\right| \\
&\leq \left|\mathbb{E}_{(x,y)\sim\mathcal{D}}\mathbb{E}_{a\sim k(x)}[\ell(a,y) - \hat{\ell}(a,y)]\right| + \left|\mathbb{E}_{(x,y)\sim\mathcal{D}}\mathbb{E}_{a\sim k(x)}[\hat{\ell}(a,y) - f_p(x,a,\ell)]\right| \\
&\leq \frac{\epsilon}{2} + \frac{\epsilon}{2} = \epsilon,
\end{aligned}$$

where the last inequality holds because $s$ gives $(\dim(\mathcal{H}), \lambda, \epsilon/2)$-uniform approximations to $\mathcal{L}_\phi$ and $f_p$ is $(\hat{\mathcal{L}}_\phi, \mathcal{K}, \epsilon/2)$-decision calibrated. $\square$

# I  PROOF OF THE LOWER BOUND IN SECTION 3

As our first step in the proof for Theorem 3.1, we derive an equivalent definition of decision calibration error for binary actions.

**Lemma I.1.** *For linear loss function class and $|\mathcal{A}| = 2$, when loss estimator $f_p$ is induced by some predictor $p$, we have*

$$\text{decCE}_{\mathcal{L}_{\text{LIN}},\mathcal{K}_{\mathcal{L}_{\text{LIN}}}}(f_p,\mathcal{D}) = \sup_{r\in\mathbb{R}^d}\|\mathbb{E}(y-p(x))\cdot\mathbf{1}(\langle r,p(x)\rangle > 0)\|_2 + \|\mathbb{E}(y-p(x))\cdot\mathbf{1}(\langle r,p(x)\rangle \leq 0)\|_2.$$

*Proof.* From the definition of $\text{decCE}_{\mathcal{L},\mathcal{K}}(f,\mathcal{D})$, we have

$$\begin{aligned}
&\text{decCE}_{\mathcal{L}_{\text{LIN}},\mathcal{K}_{\mathcal{L}_{\text{LIN}}}}(f_p,\mathcal{D}) \\
&= \sup_{\ell\in\mathcal{L}_{\text{LIN}}, k\in\mathcal{K}_{\mathcal{L}_{\text{LIN}}}}\left|\mathbb{E}_{(x,y)\sim\mathcal{D}}\mathbb{E}_{a\sim k(x)}[\ell(a,y)] - \mathbb{E}_{(x,y)\sim\mathcal{D}}\mathbb{E}_{a\sim k(x)}[f_p(x,a,\ell)]\right| \\
&= \sup_{\ell\in\mathcal{L}_{\text{LIN}}, k\in\mathcal{K}_{\mathcal{L}_{\text{LIN}}}}\left|\mathbb{E}_{(x,y)\sim\mathcal{D}}\mathbb{E}_{a\sim k(x)}[\langle r_\ell(a),y\rangle] - \mathbb{E}_{(x,y)\sim\mathcal{D}}\mathbb{E}_{a\sim k(x)}[\langle r_\ell(a),p(x)\rangle]\right| \\
&= \sup_{\ell\in\mathcal{L}_{\text{LIN}}, k\in\mathcal{K}_{\mathcal{L}_{\text{LIN}}}}\left|\mathbb{E}_{(x,y)\sim\mathcal{D}}\mathbb{E}_{a\sim k(x)}[\langle r_\ell(a),y-p(x)\rangle]\right| \\
&= \sup_{\ell,\ell'\in\hat{\mathcal{L}}_{\text{LIN}}}|\mathbb{E}_{(x,y)\sim\mathcal{D}}[\mathbf{1}(\langle r_{\ell'}(a_1) - r_{\ell'}(a_2), p(x)\rangle > 0)\langle r_\ell(a_2), y-p(x)\rangle \\
&\quad + \mathbf{1}(\langle r_{\ell'}(a_1) - r_{\ell'}(a_2), p(x)\rangle \leq 0)[\langle r_\ell(a_1), y-p(x)\rangle]| \\
&= \sup_{r\in\mathbb{R}^d,\ell'\in\mathcal{L}_{\text{LIN}}}|\mathbb{E}_{(x,y)\sim\mathcal{D}}[\mathbf{1}(\langle r,p(x)\rangle > 0)\langle r_\ell(a_2), y-p(x)\rangle \\
&\quad + \mathbf{1}(\langle r,p(x)\rangle \leq 0)[\langle r_\ell(a_1), y-p(x)\rangle]| \\
&= \sup_{r\in\mathbb{R}^d}\|\mathbb{E}(y-p(x))\cdot\mathbf{1}(\langle r,p(x)\rangle > 0)\|_2 + \|\mathbb{E}(y-p(x))\cdot\mathbf{1}(\langle r,p(x)\rangle \leq 0)\|_2.
\end{aligned}$$

The first 3 equations is from definition and simple algebra, the fourth equation holds from the definition of optiml decision rule, and the last line holds by Cauchy–Schwarz inequality. $\qquad\square$

Now we give the proof idea for Theorem 3.1. At a high level, our lower bound follows a template similar to that used in the lower bound for high-dimensional full calibration presented in Gopalan et al. (2024a), which investigates the sample complexity required to verify full calibration. To prove Theorem 3.1, we start with a set $V = \{v_1, v_2, \cdots, v_d\} \subset \mathcal{Y}$ where $v_i = \frac{1}{2}e_i$ is half of the unit vector with the $i$-th coordinate being $1/2$ and all other coordinates take value $0$. $V$ can be shattered by the function class $\mathcal{H} = \{h : h(v) = \text{sign}\langle r, v\rangle, r \in \mathbb{R}^d\}$. Formally, a set $S \subseteq X$ is said to be *shattered* by $\mathcal{H}$ if for every function $f : S \to \{-1, 1\}$ there exists a hypothesis $h \in \mathcal{H}$ such that $\forall x \in S, h(x) = f(x)$.

**Lemma I.2** (Shattering). *$V$ can be shattered by the function class $\mathcal{H} = \{h | h(v) = \text{sign}(\langle r, v\rangle), \|r\|_2 \le 1\}$.*

*Proof.* Let $r = (r^{(1)}, ..., r^{(d)}) \in \{-\frac{1}{\sqrt{d}}, \frac{1}{\sqrt{d}}\}^d$. We have $\text{sign}(\langle r, \frac{1}{2}e_i\rangle) = \text{sign}(r^{(i)})$. Therefore, $h(v)$ can arbitrarily takes value in $\{-1, 1\}$ at each point $v_i \in V$, which means $V$ can be shattered by $\mathcal{H}$. $\qquad\square$

Next we use $V$ to construct candidate distributions with large decision calibration error.

**Lemma I.3.** *For any $\sigma = (\sigma^{(1)}, ..., \sigma^{(d)}) \in \{-\frac{1}{\sqrt{d}}, \frac{1}{\sqrt{d}}\}^d$, Let $h_\sigma$ be the function that for any $i \in [d]$, it holds that $h_\sigma(v_i) = v_i + \epsilon \cdot \text{sign}(\sigma_i)e_1$. Consider a distribution $\mathcal{D}_\sigma$ and predictor $p$, such that $p(x)$ is distributed uniformly over $V$ and $y = h_\sigma(p(x))$, then it holds that $\text{decCE}_{\mathcal{L}_{\text{LIN}}, \mathcal{K}_{\mathcal{L}_{\text{LIN}}}}(f_p, \mathcal{D}_\sigma) \ge \epsilon$.*

*Proof.* Consider $r = \sigma$, we use $l = \sum_{i=1}^d \mathbf{1}(\sigma_i) > 0$ to denote the number of positive coordinates in $\sigma$. We have

$$
\begin{aligned}
\text{decCE}_{\mathcal{L}_{\text{LIN}}, \mathcal{K}_{\mathcal{L}_{\text{LIN}}}}(f_p, \mathcal{D}) &= \sup_{r \in \mathbb{R}^d} \|\mathbb{E}(y - p(x)) \cdot \mathbf{1}(\langle r, p(x)\rangle > 0)\|_2 \\
&\quad + \|\mathbb{E}(y - p(x)) \cdot \mathbf{1}(\langle r, p(x)\rangle \le 0)\|_2 \\
&\ge \|\mathbb{E}[(y - p(x))]\mathbf{1}(\langle \sigma, v\rangle > 0)\|_2 + \|\mathbb{E}[(y - p(x))]\mathbf{1}(\langle \sigma, v\rangle \le 0)\|_2 \\
&= \left\|\frac{1}{d}\sum_{i=1}^d \mathbf{1}(\sigma_i > 0)\epsilon\,\text{sign}(\sigma_i)e_1\right\|_2 + \left\|\frac{1}{d}\sum_{i=1}^d \mathbf{1}(\sigma_i < 0)\epsilon\,\text{sign}(\sigma_i)e_1\right\|_2 \\
&= \frac{l\epsilon}{d} + \frac{(d-l)\epsilon}{d} \\
&= \epsilon.
\end{aligned}
$$
$$(6)$$
$\square$

We now construct two nearly indistinguishable distributions over $n$ data points of prediction-outcome pairs $(p(x), y)$, denoted by $\mathcal{D}_1, \mathcal{D}_2 \in \Delta((\mathcal{Y} \times \mathcal{Y})^n)$. The first distribution $\mathcal{D}_1$ is such that the predictor $p$ is perfectly decision calibrated, while the second distribution $\mathcal{D}_2$ is a mixture over distributions where $p$ incurs a decision calibration error of $\epsilon$. The goal is to show that telling which of $\mathcal{D}_1$ and $\mathcal{D}_2$ generates the observations requires a number of samples $\Omega(\sqrt{d})$.

Let $A$ be an algorithm that receives $n$ samples $(p(x_1), y_1), ((p(x_2), y_2), \ldots, (p(x_n), y_n) \in \mathcal{Y}^2$ and outputs either "accept" or "reject." Define the joint distribution $\mathcal{D}_1 \in \Delta((\mathcal{Y} \times \mathcal{Y})^n)$ as follows: each $p(x_i)$ is drawn independently and uniformly from a finite set $V$, and each corresponding $y_i$ is independently drawn as $y_i = p(x_i) \pm \epsilon e_1$, where the sign is chosen uniformly at random. Let $p_1$ denote the probability that algorithm $A$ accepts $p$ on samples follow $\mathcal{D}_1$.

Next, define the joint distribution $\mathcal{D}_2 \in \Delta((\mathcal{Y} \times \mathcal{Y})^n)$ as follows: first, uniformly sample a perturbation vector $\sigma \in \{-\frac{1}{\sqrt{d}}, \frac{1}{\sqrt{d}}\}^d$, and then sample each $p(x_i)$ independently and uniformly from $V$. For each $i$, set $y_i = h_\sigma(p(x_i))$, where $h_\sigma$ is a fixed perturbation function defined by $\sigma$.

Intuitively, these two distributions are nearly identical. As long as all predictions $p(x_1), \ldots, p(x_n)$ are distinct, the behavior of $\mathcal{D}_1$ and $\mathcal{D}_2$ is almost indistinguishable. The key difference arises when

two data points share the same prediction value $v = p(x_i) = p(x_j)$: in $\mathcal{D}_1$, the outcomes $y_i$ and $y_j$ may differ due to independent noise, while in $\mathcal{D}_2$, they are always the same because the mapping $h_\sigma$ is fixed once $\sigma$ is sampled.

We now formalize this intuition in the following statement.

**Lemma I.4.** *Let $p_1$ be the probability that $A$ accepts when the data $((p(x_1), y_1), ..., (p(x_n), y_n)) \sim \mathcal{D}_1$, and Let $p_2$ be the probability that $A$ accepts when the data $((p(x_1), y_1), ..., (p(x_n), y_n)) \sim \mathcal{D}_2$. Here, the randomness comes from both the inherent randomness in $A$ and the data. Then, it holds that $|p_1 - p_2| \leq O(n^2/d)$.*

*Proof.* Without loss of generality we assume that $n < |V| = d$. For proving the lemma, we introduce another joint distribution over the $n$ data points, where we first draw $p(x_1), ...p(x_n)$ uniformly without replacement from $V$, and then for any $i$, we independently draw $y_i = p(x_i) \pm \epsilon e_i$ with both probabilities $1/2$. We use $p_3$ to denote the probability of $A$ accept if the data points follow this joint distribution $\mathcal{D}_3$.

Also, when we draw all $p(x_i)$ independently uniformly with replacement, we use $E$ to denote the event that $p(x_1), ..., p(x_n)$ turn out to be distinct. We have

$$\Pr[E] = (1 - 1/|V|)...(1 - (n-1)/|V|) \geq 1 - O(n^2/d). \tag{7}$$

For both joint distributions, conditioned on the event $E$, the probability that $A$ will accept is exactly $p_3$. Then, we have

$$\Pr[E] \cdot p_3 \leq p_1 \leq \Pr[E] \cdot p_3 + (1 - \Pr[E]).$$

We also have

$$\Pr[E] \cdot p_3 \leq p_2 \leq \Pr[E] \cdot p_3 + (1 - \Pr[E]).$$

Therefore, we have

$$|p_1 - p_2| \leq 1 - \Pr[E] \leq O(n^2/d). \tag{8}$$

$\square$

Now we are able to prove Theorem 3.1.

*Proof of Theorem 3.1.* Consider the case of $\mathcal{D}_1$, it can be viewed the data points are drawn independently from a distribution $\mathcal{D}$, where $p(x_i)$ is drawn uniformly from $V$, and $y = p(x) \pm \epsilon e_1$ with probability both $\frac{1}{2}$. Therefore $\mathcal{D}_1$ is a joint distrbution such that the data points are drawn from a distribution such that $p$ is calibrated (and therefore decision calibrated). Therefore, we have $p_1 \geq 2/3$.

Consider the case of $\mathcal{D}_2$, it can be viewed as a mixture of distributions indexed by $\sigma$, where for each distribution, the data points are drawn independently from a distribution $\mathcal{D}_\sigma$, where $p(x)$ is drawn uniformly from $V$, and $y = h_\sigma(p(x))$. The distribution is a mixture where $\sigma$ is drawn uniformly. Therefore,

$$p_2 = \frac{1}{2^d} \sum_{\sigma \in \{-1, +1\}^d} \Pr[A \text{ accepts } \mathcal{D}_\sigma]. \tag{9}$$

As a result, from Lemma I.3, we have

$$p_2 \leq \frac{1}{2^d} \sum_{\sigma \in \{-1, +1\}^d} 1/3 = 1/3.$$

By Lemma I.4, we know $n \geq \Omega(\sqrt{d})$. $\square$

## J    PROOFS IN SECTION 4

**Lemma 4.1.** *For a loss estimator $f_p$ derived from the predictor $p$, it is $(\mathcal{L}_\mathcal{H}, \tilde{\mathcal{K}}_{\mathcal{L}_\mathcal{H}}, \epsilon)$-decision calibrated if and only if*

$$\sup_{\ell, \ell' \in \mathcal{L}_\mathcal{H}} \left| \mathbb{E}_{(x,y) \sim \mathcal{D}} \left[ \sum_{a=1}^{|\mathcal{A}|} \langle r_\ell(a), \phi(y) - p(x) \rangle \tilde{k}_{f_p, \ell'}(x, a) \right] \right| \leq \epsilon. \tag{3}$$

*Proof.* By reproducing property, for any $\ell, \ell' \in \mathcal{L}_\mathcal{H}$, we have

$$
\left| \mathbb{E}_{(x,y)\sim\mathcal{D}} \mathbb{E}_{a\sim\tilde{k}_{f_p,\ell'}(x)} [\ell(a,y)] - \mathbb{E}_{(x,y)\sim\mathcal{D}} \mathbb{E}_{a\sim\tilde{k}_{f_p,\ell'}(x)} [f_p(x,a,\ell)] \right|
$$

$$
= \left| \mathbb{E}_{(x,y)\sim\mathcal{D}} \left[ \sum_{a=1}^{|\mathcal{A}|} \ell(a,y)\tilde{k}_{f_p,\ell'}(x,a) \right] - \mathbb{E}_{(x,y)\sim\mathcal{D}} [\sum_{a=1}^{|\mathcal{A}|} f_p(x,a,\ell)\tilde{k}_{f_p,\ell'}(x,a)] \right|
$$

$$
= \left| \mathbb{E}_{(x,y)\sim\mathcal{D}} \left[ \sum_{a=1}^{|\mathcal{A}|} (\ell(a,y) - f_p(x,a,\ell))\tilde{k}_{f_p,\ell'}(x,a) \right] \right|
$$

$$
= \left| \mathbb{E}_{(x,y)\sim\mathcal{D}} \left[ \sum_{a=1}^{|\mathcal{A}|} (\langle r_\ell(a), \phi(y)\rangle - \langle r_\ell(a), p(x)\rangle)\tilde{k}_{f_p,\ell'}(x,a) \right] \right|
$$

$$
= \left| \mathbb{E}_{(x,y)\sim\mathcal{D}} \left[ \sum_{a=1}^{|\mathcal{A}|} \langle r_\ell(a), \phi(y) - p(x)\rangle\tilde{k}_{f_p,\ell'}(x,a) \right] \right|.
$$

□

**Theorem 4.2** (ERM as Auditing Algorithm). *Let $D = \{(x_1,y_1),...,(x_n,y_n)\}$ be the dataset that each data point is drawn i.i.d. from $\mathcal{D}$, given any predictor $p : \mathcal{X} \to \mathcal{H}$, the ERM algorithm that outputs*

$$
\hat{\ell}, \hat{\ell}' \leftarrow \arg\max_{\ell,\ell'} \frac{1}{n} \sum_{i=1}^{n} L_{\text{DecCal}}(\ell, \ell', x_i, y_i),
$$

*when $n \geq \tilde{O}(|\mathcal{A}|^3 \beta^4 R_1^6 R_2^6 \epsilon^{-2})$, ERM algorithm is an $\epsilon$-auditor.*

*Proof.* This follows directly from Theorem 4.1, because when $n \geq \tilde{O}(\frac{|\mathcal{A}|^3 \beta^4 R_1^6 R_2^6}{\epsilon^2})$, we have

$$
\sup_{\ell,\ell'\in\mathcal{L}_\mathcal{H}} \left| \mathbb{E}_{(x,y)\sim\mathcal{D}} \left[ \sum_{a=1}^{|\mathcal{A}|} \langle r_\ell(a), \phi(y) - p(x)\rangle\tilde{k}_{f_p,\ell'}(x,a) \right] \right.
$$

$$
\left. - \mathbb{E}_{(x,y)\sim D} \left[ \sum_{a=1}^{|\mathcal{A}|} \langle r_\ell(a), \phi(y) - p(x)\rangle\tilde{k}_{f_p,\ell'}(x,a) \right] \right| \leq \epsilon/2.
$$

From the definition of $L_{\text{DecCal}}$, we know that

$$
\frac{1}{n} \sum_{i=1}^{n} L_{\text{DecCal}}(\ell, \ell', x_i, y_i) = \mathbb{E}_{(x,y)\sim D} \left[ \sum_{a=1}^{|\mathcal{A}|} \langle r_\ell(a), \phi(y) - p(x)\rangle\tilde{k}_{f_p,\ell'}(x,a) \right].
$$

Here, we can remove the absolute value since $\mathcal{L}$ is defined as the ball $\mathcal{L}_\mathcal{H} = \{\ell : \forall a, \ell(a,\cdot) \in \mathcal{H}, \|\ell(a,\cdot)\|_\mathcal{H} \leq R_1\}$, which is symmetric by construction.

By triangle inequality, when $\text{decCE}(f_p, \mathcal{D}) \geq \epsilon$, we have

$$
\left| \mathbb{E}_{(x,y)\sim\mathcal{D}} \left[ \sum_{a=1}^{|\mathcal{A}|} \langle r_{\hat{\ell}}(a), \phi(y) - p(x)\rangle\tilde{k}_{f_p,\hat{\ell}'}(x,a) \right] \right| \geq \epsilon/2.
$$

□

## K   UNIFORM CONVERGENCE FOR AUDITING DECISION CALIBRATION WITH SMOOTHED OPTIMAL DECISION RULE

Now we introduce the finite sample analysis for decision calibration under smooth optimal decision rule.

**Theorem K.1.** *Let $D = (x_1, y_1), ..., (x_n, y_n)$ be the dataset that each data point is drawn i.i.d. from $\mathcal{D}$, with probability at least $1 - \delta$ we have that*

$$
\sup_{\ell, \ell' \in \mathcal{L}_{\mathcal{H}}} \left| \mathbb{E}_{(x,y) \sim \mathcal{D}} \left[ \sum_{a=1}^{|\mathcal{A}|} \langle r_\ell(a), \phi(y) - p(x) \rangle \tilde{k}_{f_p, \ell'}(x, a) \right] \right.
$$

$$
\left. - \mathbb{E}_{(x,y) \sim D} \left[ \sum_{a=1}^{|\mathcal{A}|} \langle r_\ell(a), \phi(y) - p(x) \rangle \tilde{k}_{f_p, \ell'}(x, a) \right] \right| \tag{10}
$$

$$
\leq O \left( \frac{|\mathcal{A}|^{\frac{3}{2}} \beta^2 R_1^3 R_2^3 \log(R_1 R_2 n) + \log(1/\delta)}{\sqrt{n}} \right).
$$

Note that this bound is independent of $d$, therefore holds for infinite dimension space.

To prove the theorem, recall that we define the function class $\mathcal{G}$, where each element is a function parameterized by the loss function $\ell$ and $\ell'$. The function takes a data point as input and output the loss they the agent receives when they respond based on $\ell'$ and their true loss to be $\ell$. In detail, we have

$$
g_{l,l'}(p(x), \phi(y)) : = \sum_{a=1}^{|\mathcal{A}|} \langle r_\ell(a), \phi(y) - p(x) \rangle \tilde{k}_{f_p, \ell'}(x, a)
$$

$$
= \sum_{a=1}^{|\mathcal{A}|} \langle r_\ell(a), \phi(y) - p(x) \rangle \frac{e^{-\beta \langle \ell'_a, p(x) \rangle}}{\sum_{a'=1}^{|\mathcal{A}|} e^{-\beta \langle \ell'_{a'}, p(x) \rangle}}.
$$

Now we can show that, the difference between $g_{\ell^1, \ell^{1'}}(p(x), \phi(y))$ and $g_{\ell^2, \ell^{2'}}(p(x), \phi(y))$ is small when $\ell^1 \approx \ell^{1'}$ and $\ell^2 \approx \ell^{2'}$.

**Lemma K.1.** *Consider the vector softmax function* $\text{softmax}(z)_i = \frac{e^{-\beta z_i}}{\sum_{j=1}^{|\mathcal{A}|} e^{-\beta z_j}}$ *for each coordinate $i \in [|\mathcal{A}|]$ and $z \in \mathbb{R}^{|\mathcal{A}|}$, then we have*

$$
\|\text{softmax}(z) - \text{softmax}(z')\|_1 \leq \sqrt{2}\beta \|z - z'\|_2.
$$

*Proof.* Then by mean value theorem, we know that

$$
\text{softmax}(z) - \text{softmax}(z') = \int_{t=0}^{1} \nabla \text{softmax}(z' + (z - z')t)(z - z')dt.
$$

By taking the $\ell_1$ norm, we have

$$
\|\text{softmax}(z) - \text{softmax}(z')\|_1 \leq \int_{t=0}^{1} \|\nabla \text{softmax}(z' + (z - z')t)(z - z')\|_1 dt.
$$

Let $z_t = z' + (z - z')t$ and $p_t = \text{softmax}(z_t)$. We have $A := \nabla \text{softmax}(z_t) = -\beta(diag(p_t) - p_t p_t^T)$. Then we have

$$
\|A(z - z')\|_1 = \sum_{i=1}^{|\mathcal{A}|} \left| \sum_{j=1}^{|\mathcal{A}|} a_{ij}(z_j - z'_j) \right|
$$

$$
\leq \sum_{i=1}^{|\mathcal{A}|} \sqrt{\sum_{j=1}^{|\mathcal{A}|} a_{ij}^2} \|z - z'\|_2
$$

$$
= \sum_{i=1}^{|\mathcal{A}|} \beta \sqrt{(p_i - p_i^2)^2 + p_i^2 \sum_{j \neq i} p_j^2} \|z - z'\|_2
$$

$$
\leq \sum_{i=1}^{|\mathcal{A}|} \beta p_i \sqrt{(1 - p_i)^2 + 1} \|z - z'\|_2
$$

$$\leq \sum_{i=1}^{|\mathcal{A}|} \sqrt{2}\beta p_i \|z - z'\|_2$$

$$= \sqrt{2}\beta \|z - z'\|_2.$$

□

**Lemma K.2.** *Let* $g(z, w) := \sum_{i=1}^{|\mathcal{A}|} \frac{e^{-\beta z_i}}{\sum_{j=1}^{|\mathcal{A}|} e^{-\beta z_j}} w_i$ *for any* $z, w \in \mathbb{R}^{|\mathcal{A}|}$. *If* $\|w\|_\infty \leq 4R_1 R_2$, *we have* $|g(z, w) - g(z', w')| \leq 4\sqrt{2}R_1 R_2 \|z - z'\|_2 + \|w - w'\|_2$.

*Proof.*

$$|g(z, w) - g(z', w')|$$

$$= \left| \sum_{i=1}^{|\mathcal{A}|} \frac{e^{-\beta z_i}}{\sum_{j=1}^{|\mathcal{A}|} e^{-\beta z_j}} w_i - \sum_{i=1}^{|\mathcal{A}|} \frac{e^{-\beta z_i'}}{\sum_{j=1}^{|\mathcal{A}|} e^{-\beta z_j'}} w_i' \right|$$

$$= \left| \left( \sum_{i=1}^{|\mathcal{A}|} \frac{e^{-\beta z_i}}{\sum_{j=1}^{|\mathcal{A}|} e^{-\beta z_j}} w_i - \sum_{i=1}^{|\mathcal{A}|} \frac{e^{-\beta z_i}}{\sum_{j=1}^{|\mathcal{A}|} e^{-\beta z_j}} w_i' \right) + \left( \sum_{i=1}^{|\mathcal{A}|} \frac{e^{-\beta z_i}}{\sum_{j=1}^{|\mathcal{A}|} e^{-\beta z_j}} w_i' - \sum_{i=1}^{|\mathcal{A}|} \frac{e^{-\beta z_i'}}{\sum_{j=1}^{|\mathcal{A}|} e^{-\beta z_j'}} w_i' \right) \right|$$

$$\leq \left| \sum_{i=1}^{|\mathcal{A}|} \frac{e^{-\beta z_i}}{\sum_{j=1}^{|\mathcal{A}|} e^{-\beta z_j}} w_i - \sum_{i=1}^{|\mathcal{A}|} \frac{e^{-\beta z_i}}{\sum_{j=1}^{|\mathcal{A}|} e^{-\beta z_j}} w_i' \right| + \left| \sum_{i=1}^{|\mathcal{A}|} \frac{e^{-\beta z_i}}{\sum_{j=1}^{|\mathcal{A}|} e^{-\beta z_j}} w_i' - \sum_{i=1}^{|\mathcal{A}|} \frac{e^{-\beta z_i'}}{\sum_{j=1}^{|\mathcal{A}|} e^{-\beta z_j'}} w_i' \right|. \tag{11}$$

We first bound the first term. We have

$$\left| \sum_{i=1}^{|\mathcal{A}|} \frac{e^{-\beta z_i}}{\sum_{j=1}^{|\mathcal{A}|} e^{-\beta z_j}} w_i - \sum_{i=1}^{|\mathcal{A}|} \frac{e^{-\beta z_i}}{\sum_{j=1}^{|\mathcal{A}|} e^{-\beta z_j}} w_i' \right| = \left| \sum_{i=1}^{|\mathcal{A}|} \frac{e^{-\beta z_i}}{\sum_{j=1}^{|\mathcal{A}|} e^{-\beta z_j}} (w_i - w_i') \right| \tag{12}$$

$$\leq \|w - w'\|_\infty \leq \|w - w'\|_2.$$

Next, we are going to bound the second term. We have

$$\left| \sum_{i=1}^{|\mathcal{A}|} \frac{e^{-\beta z_i}}{\sum_{j=1}^{|\mathcal{A}|} e^{-\beta z_j}} w_i' - \sum_{i=1}^{|\mathcal{A}|} \frac{e^{-\beta z_i'}}{\sum_{j=1}^{|\mathcal{A}|} e^{-\beta z_j'}} w_i' \right| = \left| \left( \sum_{i=1}^{|\mathcal{A}|} \frac{e^{-\beta z_i}}{\sum_{j=1}^{|\mathcal{A}|} e^{-\beta z_j}} - \frac{e^{-\beta z_i'}}{\sum_{j=1}^{|\mathcal{A}|} e^{-\beta z_j'}} \right) w_i' \right| \tag{13}$$

$$\leq 4R_1 R_2 \sum_{j=1}^{|\mathcal{A}|} \left| \frac{e^{-\beta z_i}}{\sum_{j=1}^{|\mathcal{A}|} e^{-\beta z_j}} - \frac{e^{-\beta z_i'}}{\sum_{j=1}^{|\mathcal{A}|} e^{-\beta z_j'}} \right|.$$

From [Lemma K.1](#), we have

$$\left| \sum_{i=1}^{|\mathcal{A}|} \frac{e^{-\beta z_i}}{\sum_{j=1}^{|\mathcal{A}|} e^{-\beta z_j}} w_i' - \sum_{i=1}^{|\mathcal{A}|} \frac{e^{-\beta z_i'}}{\sum_{j=1}^{|\mathcal{A}|} e^{-\beta z_j'}} w_i' \right| \leq 4\sqrt{2}\beta R_1 R_2 \|z - z'\|_2. \tag{14}$$

Therefore, we know

$$|g(z, w) - g(z', w')| \leq 4\sqrt{2}\beta R_1 R_2 \|z - z'\|_2 + \|w - w'\|_2.$$

□

**Lemma K.3.** *There exists a constant* $C$, *such that for any* $x, y$, *we have*

$$\left| g_{\ell^1, \ell^{1'}}(p(x), \phi(y)) - g_{\ell^2, \ell^{2'}}(p(x), \phi(y)) \right|^2$$

$$\leq C \left( \sum_{a=1}^{|\mathcal{A}|} \langle r_{\ell^{1'}}(a) - r_{\ell^{2'}}(a), p(x) \rangle^2 + \sum_{a=1}^{|\mathcal{A}|} \langle r_{\ell^1}(a) - r_{\ell^2}(a), \phi(y) - p(x) \rangle^2 \right).$$

*Proof.* Because the norm of $p(x)$ and $\phi(y)$ is bounded by $R_2$, and the norm of loss vector $r_\ell(a)$ is bounded by $R_1$, by Lemma K.2, we know

$$
\begin{aligned}
&\left| g_{\ell^1,\ell^{1'}}(p(x),\phi(y)) - g_{\ell^2,\ell^{2'}}(p(x),\phi(y)) \right| \\
&\leq 4\sqrt{2}\beta R_1 R_2 \sqrt{\sum_{a=1}^{|\mathcal{A}|} \langle r_{\ell^{1'}}(a) - r_{\ell^{2'}}(a), p(x)\rangle^2} + \sqrt{\sum_{a=1}^{|\mathcal{A}|} \langle r_{\ell^1}(a) - r_{\ell^2}(a), \phi(y) - p(x)\rangle^2}.
\end{aligned}
\tag{15}
$$

Therefore, we have

$$
\begin{aligned}
&\left| g_{\ell^1,\ell^{1'}}(p(x),\phi(y)) - g_{\ell^2,\ell^{2'}}(p(x),\phi(y)) \right|^2 \\
&\leq \left( 4\sqrt{2}\beta R_1 R_2 \sqrt{\sum_{a=1}^{|\mathcal{A}|} \langle r_{\ell^{1'}}(a) - r_{\ell^{2'}}(a), p(x)\rangle^2} + \sqrt{\sum_{a=1}^{|\mathcal{A}|} \langle r_{\ell^1}(a) - r_{\ell^2}(a), \phi(y) - p(x)\rangle^2} \right)^2 \\
&\leq 64\beta^2 R_1^2 R_2^2 \sum_{a=1}^{|\mathcal{A}|} \langle r_{\ell^{1'}}(a) - r_{\ell^{2'}}(a), p(x)\rangle^2 + 2 \sum_{a=1}^{|\mathcal{A}|} \langle r_{\ell^1}(a) - r_{\ell^2}(a), \phi(y) - p(x)\rangle^2.
\end{aligned}
\tag{16}
$$

We can set $C = \max\{64\beta^2 R_1^2 R_2^2, 2\}$ and thus the lemma is proved. $\square$

**Lemma K.4.** *Consider* $\mathcal{G} = \{g_{\ell,\ell'} | \forall a \in \mathcal{A}, \ell(a,\cdot) \in \mathcal{H}, \|\ell(a,\cdot)\|_{\mathcal{H}} \leq R_1\}$, *then we have*

$$
\log N(\mathcal{G}, L_2^{\mathcal{G}}(P_n), \epsilon) \leq O\left( \frac{|\mathcal{A}|^3 \beta^4 R_1^6 R_2^6}{\epsilon^2} \right).
$$

*Proof.* By Lemma K.3, we know

$$
\begin{aligned}
&\sum_{i=1}^{n} \left| g_{\ell^1,\ell^{1'}}(p(x_i),\phi(y_i)) - g_{\ell^2,\ell^{2'}}(p(x_i),\phi(y_i)) \right|^2 \\
&\leq C \sum_{i=1}^{n} \left( \sum_{a=1}^{|\mathcal{A}|} \langle r_{\ell^{1'}}(a) - r_{\ell^{2'}}(a), p(x_i)\rangle^2 + \sum_{a=1}^{|\mathcal{A}|} \langle r_{\ell^1}(a) - r_{\ell^2}(a), \phi(y_i) - p(x_i)\rangle^2 \right).
\end{aligned}
$$

The high level idea is to construct covers for $2|\mathcal{A}|$ Hilbert balls, then we can bound the right-hand side. Then, the Cartisan product of these covers would be a $\epsilon$ cover for the function class $\mathcal{G}$.

By Theorem D.3, Let $L_a$ be the smallest $\frac{\epsilon}{2|\mathcal{A}|C}$-cover of $\Theta_{r_\ell(a)} := \{r_\ell(a) | r_\ell(a) \in \mathcal{H}, \|r_\ell(a)\|_{\mathcal{H}} \leq R_1\}$, we have $\log |L_a| \leq O(\frac{|\mathcal{A}|^2 C^2 R_1^2 R_2^2}{\epsilon^2})$. Therefore, $L := \prod_{a \in \mathcal{A}} L_a \times \prod_{a \in \mathcal{A}} L_a$ would become a $\epsilon$-cover of $\mathcal{L}_{\mathcal{H}} \times \mathcal{L}_{\mathcal{H}}$ under the metric $L_2^{\mathcal{G}}(P_n)$. We have

$$
\log |L| = 2 \sum_{a \in \mathcal{A}} \log |L_a| = O\left( \frac{|\mathcal{A}|^3 C^2 R_1^2 R_2^2}{\epsilon^2} \right).
$$

As we know $C = \max\{64\beta^2 R_1^2 R_2^2, 2\}$, we know

$$
\log |L| = O\left( \frac{|\mathcal{A}|^3 \beta^4 R_1^6 R_2^6}{\epsilon^2} \right).
$$

$\square$

Now we prove Theorem K.1.

*Proof.* Recalling $g_{l,l'}(p(x), \phi(y)) = \sum_{a=1}^{|\mathcal{A}|} \langle r_\ell(a), \phi(y) - p(x) \rangle \tilde{k}_{f_p,\ell'}(x, a)$, we know

$$
\begin{aligned}
|g_{l,l'}(p(x), \phi(y))| &= \left| \sum_{a=1}^{|\mathcal{A}|} \langle r_\ell(a), \phi(y) - p(x) \rangle \tilde{k}_{f_p,\ell'}(x, a) \right| \\
&\leq \sum_{a=1}^{|\mathcal{A}|} \tilde{k}_{f_p,\ell'}(x, a) |\langle r_\ell(a), \phi(y) - p(x) \rangle| \\
&\leq \sum_{a=1}^{|\mathcal{A}|} \tilde{k}_{f_p,\ell'}(x, a) 2R_1 R_2 \\
&= 2R_1 R_2.
\end{aligned}
\tag{17}
$$

Therefore, when $\epsilon \geq 2R_1 R_2$, we have $\log N(\mathcal{G}, L_2^{\mathcal{G}}(P_n), \epsilon) = \log(1) = 0$ From Theorem D.4, we know that

$$
\mathcal{R}_S(\mathcal{G}) \leq 4\alpha + 12 \int_\alpha^\infty \sqrt{\frac{\log N(\mathcal{G}, L_2^{\mathcal{G}}(P_n), \epsilon)}{n}} d\epsilon = 4\alpha + 12 \int_\alpha^{2R_1 R_2} \sqrt{\frac{\log N(\mathcal{G}, L_2^{\mathcal{G}}(P_n), \epsilon)}{n}} d\epsilon.
$$

Plugging in $\log N(\mathcal{G}, L_2^{\mathcal{G}}(P_n), \epsilon) = O(\frac{|\mathcal{A}|^3 \beta^4 R_1^6 R_2^6}{\epsilon^2})$, we have

$$
\begin{aligned}
\mathcal{R}_S(\mathcal{G}) &\leq 4\alpha + O\left( \frac{|\mathcal{A}|^{\frac{3}{2}} \beta^2 R_1^3 R_2^3}{\sqrt{n}} \right) \int_\alpha^{2R_1 R_2} \frac{1}{\epsilon} d\epsilon \\
&= 4\alpha + O\left( \frac{|\mathcal{A}|^{\frac{3}{2}}] \beta^2 R_1^3 R_2^3}{\sqrt{n}} \right) (\log 2R_1 R_2 - \log \alpha).
\end{aligned}
\tag{18}
$$

Without loss of generality we set $\alpha = \frac{|\mathcal{A}|^{\frac{3}{2}} \beta^2 R_1^3 R_2^3}{\sqrt{n}}$. If $\frac{|\mathcal{A}|^{\frac{3}{2}} \beta^2 R_1^3 R_2^3}{\sqrt{n}} > 2R_1 R_2$, we have $\mathcal{R}_S(\mathcal{G}) \leq 4\alpha \leq O(\frac{|\mathcal{A}|^{\frac{3}{2}} \beta^2 R_1^3 R_2^3}{\sqrt{n}})$. If $\frac{|\mathcal{A}|^{\frac{3}{2}} \beta^2 R_1^3 R_2^3}{\sqrt{n}} \leq 2R_1 R_2$,

$$
\begin{aligned}
\mathcal{R}_S(\mathcal{G}) &\leq O\left( \frac{|\mathcal{A}|^{\frac{3}{2}} \beta^2 R_1^3 R_2^3}{\sqrt{n}} \right) + O\left( \frac{|\mathcal{A}|^{\frac{3}{2}} \beta^2 R_1^3 R_2^3}{\sqrt{n}} \right) \left( \log 2R_1 R_2 - \log \left( O\left( \frac{|\mathcal{A}|^{\frac{3}{2}} \beta^2 R_1^3 R_2^3}{\sqrt{n}} \right) \right) \right) \\
&= O\left( \frac{|\mathcal{A}|^{\frac{3}{2}} \beta^2 R_1^3 R_2^3 \log(R_1 R_2 n)}{\sqrt{n}} \right).
\end{aligned}
$$

Then by Theorem D.2, we know with at least probability $1 - \delta/2$

$$
\sup_{g \in \mathcal{G}} \left[ \frac{1}{n} \sum_{i=1}^n g(p(x_i), \phi(y_i)) - \mathbb{E}_{(x,y) \sim \mathcal{D}}[g(p(x), \phi(y))] \right]
$$

$$
\leq 2\mathbb{E}_{S \sim \mathcal{D}^n}[\mathcal{R}_S(\mathcal{G})] + \sqrt{\frac{\log(2/\delta)}{2n}}
$$

$$
\leq O\left( \frac{|\mathcal{A}|^{\frac{3}{2}} \beta^2 R_1^3 R_2^3 \log(R_1 R_2 n) + \log(1/\delta)}{\sqrt{n}} \right).
$$

Similarly we can bound the Rademacher Complexity of function class $-\mathcal{G} := \{-g_{\ell,\ell'} \mid \ell, \ell' \in \mathcal{L}_{\mathcal{H}}\}$, and have with probability $1 - \delta/2$

$$
\sup_{g \in \mathcal{G}} \left[ \mathbb{E}_{(x,y) \sim \mathcal{D}}[g(p(x), \phi(y))] - \frac{1}{n} \sum_{i=1}^n g(p(x_i), \phi(y_i)) \right]
$$

$$
\leq O\left( \frac{|\mathcal{A}|^{\frac{3}{2}} \beta^2 R_1^3 R_2^3 \log(R_1 R_2 n) + \log(1/\delta)}{\sqrt{n}} \right).
$$

Putting them together, we have

$$\left| \sup_{g \in \mathcal{G}} \left[ \frac{1}{n} \sum_{i=1}^{n} g(p(x_i), \phi(y_i)) - \mathbb{E}_{(x,y) \sim \mathcal{D}}[g(p(x), \phi(y))] \right] \right|$$

$$\leq O\left( \frac{|\mathcal{A}|^{\frac{3}{2}} \beta^2 R_1^3 R_2^3 \log(R_1 R_2 n) + \log(1/\delta)}{\sqrt{n}} \right)$$

with probability $1 - \delta$. $\qquad \square$

## L  PROOFS OF SECTION 5

**Theorem 5.1.** *Given any initial predictor $p_0$ and tolerance $\epsilon$, Algorithm 1 terminates in $T = O(\frac{R_1^2 R_2^2}{\epsilon^2})$ iterations. Given $\tilde{O}(\frac{|\mathcal{A}|^3 R_1^8 R_2^8}{\epsilon^4})$ samples, with probability $1 - \delta$, Algorithm 1 outputs a predictor $p_T$ such that $f_{p_T}$ is $(\mathcal{L}_{\mathcal{H}}, \tilde{\mathcal{K}}_{\mathcal{L}_{\mathcal{H}}}, \epsilon)$-decision calibrated and $\mathbb{E}[\|p_T(x) - \phi(y)\|_{\mathcal{H}}^2] \leq \mathbb{E}[\|p_0(x) - \phi(y)\|_{\mathcal{H}}^2]$.*

*Proof.* If the algorithm does not terminate at round $t$, we have

$$\sup_{\ell, \ell'} \mathbb{E}\left[ \left\langle \sum_{a=1}^{|\mathcal{A}|} r_\ell(a) \tilde{k}_{\ell'}(x, a), \phi(y) - p(x) \right\rangle \right] > \epsilon.$$

By uniform convergence property we can find $\ell_t, \ell'_t$ such that

$$\hat{\mathbb{E}}\left[ \left\langle \sum_{a=1}^{|\mathcal{A}|} r_{\ell_t}(a) \tilde{k}_{\ell'_t}(x, a), \phi(y) - p(x) \right\rangle \right] > 3\epsilon/4.$$

Let $r_{\ell_t^*}(a) = R_1 \hat{\mathbb{E}}[(\phi(y) - p(x)) \tilde{k}_{\ell'_t}(x, a)] / \|\hat{\mathbb{E}}[(\phi(y) - p(x)) \tilde{k}_{\ell'_t}(x, a)]\|$, by Cauchy inequality we have

$$\sum_{a=1}^{|\mathcal{A}|} \left\langle r_{\ell_t^*}(a), \hat{\mathbb{E}}[(\phi(y) - p(x)) \tilde{k}_{\ell'_t}(x, a)] \right\rangle \geq \hat{\mathbb{E}}\left[ \left\langle \sum_{a=1}^{|\mathcal{A}|} r_{\ell_t}(a) \tilde{k}_{\ell'_t}(x, a), \phi(y) - p(x) \right\rangle \right] > 3\epsilon/4.$$

Again by uniform convergence,

$$\mathbb{E}\left[ \left\langle \sum_{a=1}^{|\mathcal{A}|} r_{\ell_t^*}(a) \tilde{k}_{\ell'_t}(x, a), \phi(y) - p(x) \right\rangle \right] > \epsilon/2.$$

$$\mathbb{E}\left[ \|p_t(x) - \phi(y)\|_{\mathcal{H}}^2 \right] - \mathbb{E}\left[ \|p_{t+1}(x) - \phi(y)\|_{\mathcal{H}}^2 \right]$$

$$\geq \mathbb{E}\left[ \|p_t(x) - \phi(y)\|_2^2 \right] - \mathbb{E}\left[ \left\| p_t(x) - \phi(y) + \sum_{a=1}^{|\mathcal{A}|} d_{ta} \tilde{k}_{\ell'_t}(x, a) \right\|_{\mathcal{H}}^2 \right]$$

$$= \sum_{a=1}^{|\mathcal{A}|} \frac{2\eta R_1}{\|\hat{\mathbb{E}}[(\phi(y) - p(x)) \tilde{k}_{\ell'_t}(x, a)]\|} \mathbb{E}[(\phi(y) - p_t(x)) \tilde{k}_{\ell'_t}(x, a)] \hat{\mathbb{E}}[(\phi(y) - p_t(x)) \tilde{k}_{\ell'_t}(x, a)]$$

$$- \mathbb{E}\left[ \left\| \sum_{a=1}^{|\mathcal{A}|} d_{ta} \tilde{k}_{\ell'_t}(x, a) \right\|_{\mathcal{H}}^2 \right]$$

$$\geq \eta\epsilon - \mathbb{E}\left[\left\|\sum_{a=1}^{|\mathcal{A}|} d_{ta}\tilde{k}_{\ell'_t}(x,a)\right\|_{\mathcal{H}}^2\right]$$

$$\geq \eta\epsilon - \eta^2 R_1^2.$$

Setting $\eta = \frac{\epsilon}{2R_1^2}$, we have

$$\mathbb{E}\left[\|p_t(x) - \phi(y)\|_{\mathcal{H}}^2\right] - \mathbb{E}\left[\|p_{t+1}(x) - \phi(y)\|_{\mathcal{H}}^2\right] \geq \frac{\epsilon^2}{4R_1^2}.$$

Therefore the algorithm will terminate in at most $\frac{16R_1^2 R_2^2}{\epsilon^2}$ iterations because $\mathbb{E}[\|p_0(x) - \phi(y)\|_2^2] \leq (2R_2)^2 = 4R_2^2$. $\qquad\square$

**Proposition 5.1.** *For Algorithm 1, if the input predictor satisfies $p_0(x) = \sum_{i=1}^{N_0} \alpha_{0i}(x)\phi(y_{0i})$, the for any $t$, we have $p_t(x) = \sum_{i=1}^{N_t} \alpha_{ti}(x)\phi(y_{ti})$.*

*Proof.* For Algorithm 1, the update in round $t$ is $p_{t+1} = \pi_{B(R_2)}(p_t(x) + \eta \cdot w_{\ell_t,\ell'_t}(p_t(x)))$ where $w_{\ell_t,\ell'_t}$ is the patching term. By induction, it suffices to prove that $w_{\ell_t,\ell'_t}(p_t(x))$ can be explicitly represented by the linear combination of $\phi(y)$. Let $S_t = \{(x'_{ti}, y'_{ti})\}_{i=1}^{n_t}$ be the set of samples used for auditing. Then we have

$$w_{\ell_t,\ell'_t}(p_t(x)) = \sum_{a=1}^{|\mathcal{A}|} \frac{R_1 \tilde{k}_{\ell'_t}(x,a)}{\|\hat{\mathbb{E}}_{S_t}[(\phi(y) - p_t(x))\tilde{k}_{\ell'_t}(x,a)]\|_{\mathcal{H}}} \cdot \hat{\mathbb{E}}_{S_t}[(\phi(y) - p_t(x))\tilde{k}_{\ell'_t}(x,a)]$$

By induction, we have $p_t(x) = \sum_{i=1}^{N_t} \alpha_{t,i}(x)\phi(y_{t,i})$. Then we can compute the smooth optimal decision rule $\tilde{k}_{\ell'_t}$ as

$$\tilde{k}_{\ell'_t}(x,a) = \frac{e^{-\beta f_{p_t}(x,a,\ell'_t)}}{\sum_{a'\in\mathcal{A}} e^{-\beta f_{p_t}(x,a',\ell'_t)}}$$

$$= \frac{e^{-\beta \sum_{i=1}^{N_t} \alpha_{t-1,i}(x)\ell'_t(a,y_{t,i})}}{\sum_{a'\in\mathcal{A}} e^{-\beta \sum_{i=1}^{N_t} \alpha_{t,i}(x)\ell'_t(a',y_{t,i})}}.$$

Then the norm can be computed as

$$\left\|\hat{\mathbb{E}}_{S_t}[(\phi(y) - p_t(x))\tilde{k}_{\ell'_t}(x,a)]\right\|_{\mathcal{H}}^2$$

$$= \left\|\frac{1}{n_t}\sum_{i=1}^{n_t}(\phi_{y'_{ti}} - p_t(x'_{ti}))\tilde{k}_{\ell'_t}(x'_{ti},a)\right\|_{\mathcal{H}}^2$$

$$= \frac{1}{n_t^2}\sum_{i,j\in[n_t]} \tilde{k}_{\ell'_t}(x'_{ti},a)\tilde{k}_{\ell'_t}(x'_{tj},a)\left\langle \phi_{y'_{ti}} - p_t(x'_{ti})), \phi_{y'_{tj}} - p_t(x'_{tj}))\right\rangle_{\mathcal{H}}$$

$$= \frac{1}{n_t^2}\sum_{i,j\in[n_t]} \tilde{k}_{\ell'_t}(x'_{ti},a)\tilde{k}_{\ell'_t}(x'_{tj},a) \cdot \left(K(y'_{ti},y'_{tj}) - \sum_{q\in[N_t]} \alpha_{tq}(x'_{ti})K(y_{tq},y'_{tj})\right.$$

$$\left. - \sum_{q\in[N_t]} \alpha_{tq}(x'_{tj})K(y_{tq},y'_{ti}) + \sum_{q,s\in[N_t]} \alpha_{tq}(x'_{ti})\alpha_{ts}(x'_{tj})K(y_{tq},y_{ts})\right).$$

Note that the empirical expectation is a linear combination of $\phi(y)$.

$$\hat{\mathbb{E}}_{S_t}[(\phi(y) - p_t(x))\tilde{k}_{\ell'_t}(x,a)]$$

$$= \frac{1}{n_t} \sum_{i \in [n_t]} (\phi(y'_{ti}) - p_t(x'_{ti})) \tilde{k}_{\ell'_t}(x'_{ti}, a)$$

$$= \frac{1}{n_t} \sum_{i \in [n_t]} \left( \phi(y'_{ti}) - \sum_{j \in [N_t]} \alpha_{tj}(x'_{ti})\phi(y_{tj}) \right) \tilde{k}_{\ell'_t}(x'_{ti}, a)$$

$$= \sum_{i \in [n_t]} \frac{\tilde{k}_{\ell'_t}(x'_{ti}, a)}{n_t} \phi(y'_{ti}) - \sum_{j \in N_t} \left( \sum_{i \in [n_t]} \frac{\alpha_{tj}(x'_{ti})\tilde{k}_{\ell'_t}(x'_{ti}, a)}{n_t} \right) \phi(y_{tj}).$$

Bringing these components together, we know the linear representation of $\Delta_t$ by $\{\phi(y'_{ti})\}_{i=1}^{n_t} \bigcup \{\phi(y_{ti})\}_{i=1}^{N_t}$ can be computed. For ease of notion, we let $\{y_{t+1,i}\}_{i=1}^{N_{t+1}}$ to be the union of two set of samples mentioned above. By induction we know that the linear representation of $p_t(x) + w_{\ell_t, \ell'_t}(p_t(x))$ can be computed. Let $p_t(x) + w_{\ell_t, \ell'_t}(p_t(x)) = \sum_{i \in [N_{t+1}]} \alpha'_{t+1,i}(x)\phi(y_{t+1,i})$.

The last thing to show is that after projection $\pi_{B(R_2)}$, the linear representation can still be computed. We have

$$\pi_{B(R_2)}(p_{t+1}(x)) = \frac{R_2}{\|p_{t+1}(x)\|_{\mathcal{H}}} \cdot p_{t+1}(x).$$

The it suffices to show that the norm is computable. We have

$$\|p_{t+1}(x)\|_{\mathcal{H}}^2 = \langle p_{t+1}(x), p_{t+1}(x) \rangle_{\mathcal{H}}$$

$$= \left\langle \sum_{i \in [N_{t+1}]} \alpha'_{t+1,i}(x)\phi(y_{t+1,i}), \sum_{i \in [N_{t+1}]} \alpha'_{t+1,i}(x)\phi(y_{t+1,i}) \right\rangle_{\mathcal{H}}$$

$$= \sum_{i,j \in [N_{t+1}]} \alpha'_{t+1,i}(x)\alpha'_{t+1,j}(x)K(y_{t+1,i}, y_{t+1,j}).$$

$\square$

## M EXTENSION OF ZHAO ET AL. (2021)'S ALGORITHM

In this section we provide an extension to Zhao et al. (2021)'s algorithm on decision calibration under smoothed optimal decision rule. Our extension also adopts a "patching"-style approach, with the patching component derived from an optimization perspective, following the intuition in their work. Consider that we find the pair of $\ell_t, \ell'_t$ in round $t$ that violates the decision calibration. If we let the patching have the following form

$$p_{t+1}(x) = p_t(x) + U\tilde{k}_{\ell'_t}(x),$$

we can *heuristically* minimize

$$L(U) := \sum_{a=1}^{|\mathcal{A}|} \left\| \mathbb{E}[(\phi(y) - p_t(x))\tilde{k}_{\ell'_t}(x,a)] \right\|^2 + \lambda\|U\|^2, \tag{19}$$

where the first term is trying to decrease the violation of decision calibration and the second term is trying to restrict the norm of $U$ so that $U$ can be efficiently approximated with samples. By simple calculation we have

$$L(U) = \sum_{a=1}^{|\mathcal{A}|} \left\| G_a - (DU^T)_a \right\|_F^2 + \lambda\|U\|_F^2$$

$$= \left\| G - DU^T \right\|_F^2 + \lambda\|U\|_F^2.$$

The optimum of the objective is $U^* = G^T(D + \lambda I)^{-1}$. Note that the optimization objective of Zhao et al. (2021) is just the first term of Eq. (19) without the second regularization term. Consequently, the optimum becomes $U^* = G^T D^{-1}$ so that the norm of $U^*$ can be unbounded because the

---

**Algorithm 2** Finite-Sample Infinite-Dimensional Adaptation of Zhao et al. (2021)

---

**Input:** The RKHS kernel $K$, current predictor $p_0 : \mathcal{X} \to \mathcal{H}$ and tolerance $\epsilon$.

1: $t = 0$
2: **while** $\exists \ell_t, \ell_t'$ such that $\mathbb{E}[\langle \sum_{a=1}^{|\mathcal{A}|} r_{\ell_t}(a)\tilde{k}_{\ell_t'}(x,a), \phi(y) - p(x)\rangle] > \epsilon$ **do**
3:    Compute $\hat{D} \in \mathbb{R}^{|\mathcal{A}| \times |\mathcal{A}|}$ where $\hat{D}_{aa'} = \hat{\mathbb{E}}[\tilde{k}_{\ell_t'}(x,a)\tilde{k}_{\ell_t'}(x,a')]$.
4:    Define $\hat{G} \in \mathbb{R}^{K \times \dim(\mathcal{H})}$ where $\hat{G}_a = \hat{\mathbb{E}}[(\phi(y) - p_t(x))\tilde{k}_{\ell_t'}(x,a)]$.
5:    Set $p_{t+1} : x \mapsto \pi_{B(R_2)}(p_t(x) + \hat{G}^T(\hat{D} + I)^{-1}\tilde{k}_{\ell_t'}(x))$.   $/\!/\pi_{B(R_2)}$ *projects onto Hilbert ball* $B(R_2)$
6: **end while**

---

(pseudo)inverse of $D$ may not have a bounded norm. Therefore, their algorithm does not have finite sample guarantee. Our regularized extension to their algorithm Algorithm 2 can fix this problem. In Algorithm 2, we choose $\lambda = 1$.

The following theorem says that Algorithm 2 is also a valid algorithm for decision calibration, but has sample complexity bounds $\tilde{O}(\frac{1}{\epsilon^6})$ worse than ours $\tilde{O}(\frac{1}{\epsilon^4})$.

**Theorem M.1.** *Given any initial predictor $p_0$ and tolerance $\epsilon$, Algorithm 2 ends in $T = O(\frac{|\mathcal{A}|R_1^2 R_2^2}{\epsilon^2})$ iterations. Given $\tilde{O}(\frac{|\mathcal{A}|^8 R_1^6 R_2^6}{\epsilon^6})$ samples, with probability $1 - \delta$, Algorithm 2 outputs a predictor $p_T$ such that $f_{p_T}$ is $(\mathcal{L}_{\mathcal{H}}, \tilde{\mathcal{K}}_{\mathcal{L}_{\mathcal{H}}}, \epsilon)$-decision calibrated and $\mathbb{E}[\|p_T(x) - \phi(y)\|_{\mathcal{H}}^2] \leq \mathbb{E}[\|p_0(x) - \phi(y)\|_{\mathcal{H}}^2]$.*

*Proof.* First we have

$$\left\|\hat{G}\right\|_F \leq 2R_2\sqrt{|\mathcal{A}|}.$$

$$\left\|(\hat{D} + I)^{-1}\right\|_F = \sqrt{\sum_{i=1}^{|\mathcal{A}|} \sigma_i((\hat{D} + I)^{-1})} \leq \sqrt{|\mathcal{A}|}.$$

$$\begin{aligned}
&\mathbb{E}\left[\|p_t(x) - \phi(y)\|_{\mathcal{H}}^2\right] - \mathbb{E}\left[\|p_{t+1}(x) - \phi(y)\|_{\mathcal{H}}^2\right] \\
=&2\mathbb{E}[(\phi(y) - p_t(x))\hat{G}^T(\hat{D} + I)^{-1}\tilde{k}_{\ell_t'}(x)] - \mathbb{E}[k_{\ell_t'}(x)^T(\hat{D} + I)^{-T}\hat{G}\hat{G}^T(\hat{D} + I)^{-1}k_{\ell_t'}(x)] \\
=&2\operatorname{Tr}(\mathbb{E}[(\phi(y) - p_t(x))\hat{G}^T(\hat{D} + I)^{-1}\tilde{k}_{\ell_t'}(x)]) - \operatorname{Tr}(\mathbb{E}[k_{\ell_t'}(x)^T(\hat{D} + I)^{-T}\hat{G}\hat{G}^T(\hat{D} + I)^{-1}k_{\ell_t'}(x)]) \\
=&2\operatorname{Tr}(\mathbb{E}[\tilde{k}_{\ell_t'}(x)(\phi(y) - p_t(x))\hat{G}^T(\hat{D} + I)^{-1}]) - \operatorname{Tr}(\mathbb{E}[k_{\ell_t'}(x)k_{\ell_t'}(x)^T(\hat{D} + I)^{-T}\hat{G}\hat{G}^T(\hat{D} + I)^{-1}]) \\
=&2\operatorname{Tr}(G\hat{G}^T(\hat{D} + I)^{-1}) - \operatorname{Tr}(D(\hat{D} + I)^{-T}\hat{G}\hat{G}^T(\hat{D} + I)^{-1}) \\
=&2\operatorname{Tr}(G\hat{G}^T(\hat{D} + I)^{-1}) - \operatorname{Tr}((D + I)(\hat{D} + I)^{-T}\hat{G}\hat{G}^T(\hat{D} + I)^{-1}) + \operatorname{Tr}((\hat{D} + I)^{-T}\hat{G}\hat{G}^T(\hat{D} + I)^{-1}) \\
\geq&2\operatorname{Tr}(G\hat{G}^T(\hat{D} + I)^{-1}) - \operatorname{Tr}((D + I)(\hat{D} + I)^{-T}\hat{G}\hat{G}^T(\hat{D} + I)^{-1}) \\
=&2\operatorname{Tr}(\hat{G}\hat{G}^T(\hat{D} + I)^{-1}) - 2\operatorname{Tr}((\hat{G} - G)\hat{G}^T(\hat{D} + I)^{-1}) \\
&- \operatorname{Tr}((\hat{D} + I)(\hat{D} + I)^{-T}\hat{G}\hat{G}^T(\hat{D} + I)^{-1}) - \operatorname{Tr}((D - \hat{D})(\hat{D} + I)^{-T}\hat{G}\hat{G}^T(\hat{D} + I)^{-1}) \\
=&\operatorname{Tr}(\hat{G}\hat{G}^T(\hat{D} + I)^{-1}) - 2\operatorname{Tr}((\hat{G} - G)\hat{G}^T(\hat{D} + I)^{-1}) - \operatorname{Tr}((D - \hat{D})(\hat{D} + I)^{-T}\hat{G}\hat{G}^T(\hat{D} + I)^{-1}) \\
\geq&\operatorname{Tr}(\hat{G}\hat{G}^T(\hat{D} + I)^{-1}) - 2\left\|\hat{G} - G\right\|_F\left\|\hat{G}^T\right\|_F\left\|(\hat{D} + I)^{-1}\right\|_F - \left\|(D - \hat{D})\right\|_F\left\|\hat{G}^T\right\|_F^2\left\|(\hat{D} + I)^{-1}\right\|_F^2 \\
\geq&\operatorname{Tr}(\hat{G}\hat{G}^T(\hat{D} + I)^{-1}) - 4R_2|\mathcal{A}|\left\|\hat{G} - G\right\|_F - 4R_2^2|\mathcal{A}|^2\left\|(D - \hat{D})\right\|_F \\
\geq&\frac{1}{2}\operatorname{Tr}(\hat{G}\hat{G}^T) - 4R_2|\mathcal{A}|\left\|\hat{G} - G\right\|_F - 4R_2^2|\mathcal{A}|^2\left\|(D - \hat{D})\right\|_F \\
\geq&O\left(\frac{\epsilon^2}{|\mathcal{A}|R_1^2}\right).
\end{aligned}$$

Therefore, the algorithm terminates in $O(|\mathcal{A}|R_1^2 R_2^2/\epsilon^2)$ rounds. By Theorem D.1, each round requires $\tilde{O}(1/\epsilon^4)$ samples to estimate $\hat{D}$ and $\hat{G}$, resulting in an overall sample complexity of $\tilde{O}(1/\epsilon^6)$. $\qquad\square$

Similarly to Proposition 5.1, we can apply the patching in Algorithm 2 implicitly.

