# OpenReview forum: "Dimension-Free Decision Calibration for Nonlinear Loss Functions"
_ICLR.cc/2026/Conference — ICLR 2026 Poster_

### Official Review · Reviewer_YoMP · 2025-10-31

**Soundness:** 4
**Presentation:** 3
**Contribution:** 3
**Rating:** 8
**Confidence:** 3

**Summary:**

The paper studies prediction for downstream decision-making via the lens of decision calibration. Decision calibration requires that predictions are calibrated with respect to a decision-making rule, which incurs loss depending on the prescribed action and label. The paper focuses on non-linear losses that admit (possibly infinite-dimensional) linear representations, e.g. in an RKHS. Prior work on decision calibration handle linear losses and losses with finite-dimensional linear representations.

The main results are:
- Under best response, verifying decision calibration requires sample complexity polynomial in the dimension of the prediction space.
- Under smooth best response, there is an auditing algorithm for losses in RKHS with sample complexity independent of the dimension.
- The paper then gives an algorithm that post-processes an initial predictor to a decision-calibrated predictor--under smooth best response and for losses in RKHS. The sample complexity is independent of the dimension and scales polynomially in the number of actions and 1/epsilon.

**Strengths:**

- I think this paper makes solid contributions to the study of calibration for decision-making. The proposed algorithm is a fairly significant extension of previous results. The authors also give a lower bound that has implications for existing calibration algorithms.
- The technical contribution is a uniform convergence argument for decision calibration under smooth best response.
- Overall the paper is well-written and the relation to previous work is clear.

**Weaknesses:**

- The idea of uniformly approximating non-linear losses comes from previous work of Gopalan et al (2024b) and Lu et al (2025) in similar contexts. The algorithmic template follows that of Gopalan et al (2022) and Zhao et al (2021) (but it was not previously clear how to handle the infinite-dimensional case).
- The uniform approximation framework is introduced but not explicitly used; even the examples in Appendix F seem to have exact representations. It would be nice to spell out the end-to-end decision calibration guarantees for losses that can be uniformly approximated and/or the examples given in Appendix F.

Minor things:
- The term “feature space” is used for both $X$ and $H$; I would suggest changing one.
- The decision rule $k$ has domain $X$, but the paper is only concerned with decision rules that are functions of the predictions (or loss estimators), right?

**Questions:**

- Why is smooth best response able to give stronger results intuitively-speaking? I think it would benefit the paper to add some intuitive explanation behind the separation, e.g. why smooth best response “fixes” the hard example of Theorem 3.1.
- Can the results under smooth best response be extended to any decision rule that is near-optimal and smooth?

---

> ### Author Response · Authors · 2025-11-24
>
> Thank you for taking the time to review our submission! Please find our replies to your questions/comments below.
>
> **W1 regarding the technical contributions:**
>
> We believe the main contribution of this paper is to address the open question of whether decision calibration can be achieved with dimension-free statistical complexity, and to characterize the settings in which this is possible. Conceptually, Gopalan et al. (2024b) and Lu et al. (2025) study related but distinct notions, omniprediction and decision swap regret, whereas our work focuses specifically on decision calibration.
>
> While the overall algorithmic template is not entirely new, we believe our method offers a clean approach that achieves dimension-free guarantees for decision calibration. Moreover, to the best of our knowledge, we provide the first finite-sample algorithm that attains decision calibration under the smoothed optimal decision rule.
>
> We agree that it is a very interesting open problem to study whether decision calibration can be achieved without relying on such iterative algorithmic frameworks which potentially yields improved sample complexity. We leave this problem for future work.
>
> **W2 regarding the uniform approximation:**
>
> Thank you for pointing this out! We will include a theorem in the final version of the paper to clarify this.
>
> **M1 regarding the terminology:**
>
> Thank you for pointing this out! We have revised the paper to use the term “context space” to denote $\mathcal{X}$ and “feature space” to denote $\mathcal{H}$.
>
>
> **M2 regarding clarification on the decision rules:**
>
> Thank you for pointing this out! Yes, the decision rule we consider in this paper only take the prediction/loss estimation as input, we use the general domain $\mathcal{X}$ just for simplifying the notation. We will add an paragraph of explanation in the final version.
>
> **Q1 regarding the intuition behind dimension-free guarantees**:
>
> We sidestep the dependence on the dimension by considering the quantal response instead of the hard best response. Intuitively, the quantal response avoids sharp, discontinuous decision boundaries. The smoothness of the quantal response allows us to project the loss vectors in any high-dimensional feature space into 1-D space by a pseudometric:
> $$
> d(r_{l_1}, r_{l_2}) = \sqrt{\mathbb{E}[\langle r_{l_1} - r_{l_2}, p(X) \rangle^2]}
> $$
> The smoothness of the quantal response implies that if the pseudometric between two loss vectors is small, their decision rule is also close. This smoothness allows our analysis to rely on the projected 1D pseudo-metric, hiding the infinite-dimensional feature space $\phi(Y)$ from the auditing algorithm.
>
>
> **Q2 regarding the extension on general decision rules**:
>
> Thank you for bringing up this question! This is a nice observation. Our result for the quantal response model extends to any response function $\tilde{k} _{f,\ell}$ that maps estimated losses to a distribution over actions, as long as two conditions are satisfied: (i) the response function gives an approximate best response, and (ii) the induced action probabilities $\tilde{k} _{f,\ell}(x,a)$ are Lipschitz in the estimated losses. In all of our proofs, we rely only on these two properties and do not use any features specific to quantal best responses. We adopt the quantal response model in the paper primarily for simplicity, and because it is one of the most commonly used behavioral models for smoothed responses. We will include this discussion in the final version of the paper.

---

### Official Review · Reviewer_F8eW · 2025-11-01

**Soundness:** 4
**Presentation:** 4
**Contribution:** 3
**Rating:** 8
**Confidence:** 4

**Summary:**

This paper studies decision calibration in high-dimensional space and nonlinear loss functions. The main results in the paper include:
* Lowerbound for deterministic best response: distinguishing decision calibration requires a sample complexity of $\Omega(\sqrt{m})$.
* Dimension-free decision calibration under smooth best response: under the quantal response model, the paper designs an algorithm that audits decision calibration, where the sample complexity is invariant of dimension.
* dimension-free audition then leads to a dimension-free algorithm for decision calibration under quantal response.

**Strengths:**

The paper addresses a well-motivated open problem. The main message in the paper is strong: under smooth best responding, decision calibration can be achieved without dependence on dimensionality. The main technical contribution in the paper is the dimension-free audition result. The algorithm design follows from existing work and the existence of the audition algorithm.

The paper is well organized. The class of functions considered here seem quite general (including nonlinear losses linearizable or uniformly approximable in an RKHS). It would be great if the authors could include more discussions on the class of loss functions admitted by the problem setup and the dependence on norm. It would also help me if there are examples not admitted in the class, and how decision calibration cannot be achieved.

**Weaknesses:**

The work builds on established techniques, yet delivers useful results.

**Questions:**

See my comment above.

---

> ### Author Response · Authors · 2025-11-24
>
> Thank you for taking the time to review our submission! Please find our replies to your questions/comments below.
>
> **Question regarding the discussion on function classes:**
>
> Thank you for the great question! In short, we believe the main advantage of our dimension-free decision calibration analysis is that it yields a much sharper dependence on $\epsilon$, avoiding the $\frac{1}{\epsilon^{\mathrm{poly}(d)}}$ dependence that becomes intractable when $\epsilon$ is small and the dimension $d$ is large Here $d$ is the dimension of the outcome space but not the space of the feature expansion.
>
> When the outcome space $\mathcal{Y} = [0,1]$, the result is quite general. If we choose the separable kernel $K(s,t) = \min(s,t) + 1$, then any function satisfying $|f(0)| \le C_1$ and $\int_0^1 (f'(x))^2 , dx \le C_2$ has a bounded RKHS norm. This already covers a broad class, as all bounded Lipschitz differentiable functions form a subset of this space. In this case, our dimension-free analysis gives a sharp $1/\epsilon^4$ dependence, whereas standard discretization would lead to a worse dependence on $\epsilon$.
>
> In $d$-dimensional outcome space (for example $\mathcal{Y} = [0,1]^d$) setting, the RKHS norm for the most expressive classes scales as $c^d$ for some constant $c$. For instance, if we take $K$ to be a Sobolev/Matérn kernel on $[0,1]^d$, then any payoff $f$ with uniformly bounded partial derivatives up to order $m>d/2$, i.e.
> $$
> \sup_{y\in[0,1]^d}\ \max_{|\alpha|\le m} \big|D^\alpha \ell(a,y)\big| \le M,
> $$
> lies in the corresponding RKHS with norm controlled by $c^d$ for some constant $c$.
> Crucially, this $c^d$ factor depends only on the geometric complexity of the function class (via $d$ and $m$) and does **not** interact with the calibration accuracy $\epsilon$. By contrast, the feature expansion in Lu et al. (2025) [1] under strict best response has dimension $\epsilon^{\mathrm{poly}(d)}$, which inevitably results in a $\frac{1}{\epsilon^{\mathrm{poly}(d)}}$ sample complexity.
>
> To conclude, we believe that adopting quantal response and following our RKHS-based analysis allows us to obtain the sharp $1/\epsilon^4$ dependence regardless of the outcome-space dimension $d$. Another important benefit is that our algorithm is hyperparameter-free since our algorithm does not require discretization, whereas the approaches of Lu et al. (2025) and Gopalan et al. (2024b) require discretization tuned to the target error $\epsilon$, which inherently worsens their $\epsilon$-dependence. Finally, we expect that techniques from adaptive data analysis may further improve the $1/\epsilon^4$ sample complexity appearing in our results.
>
> We will include a section in the final version to discuss the above points in detail.

---

### Official Review · Reviewer_zBGQ · 2025-11-01

**Soundness:** 3
**Presentation:** 2
**Contribution:** 3
**Rating:** 8
**Confidence:** 4

**Summary:**

The paper proposes improved upper and lower bounds on the problem of decision calibration in the batch setting, when faced with potentially nonlinear losses and/or high/infinite dimensional feature spaces. The contribution is roughly threefold. First, a lower bound is provided, which establishes that under the pure best response decision rule (deterministic selection of the best alternative), a poly(m) complexity in the dimension m is required just to audit for the presence/absence of decision calibration.

Second, in contrast to the first result, an efficient auditing scheme is developed for various loss classes in RKHS that has complexity independent of the ambient dimension. Third, this dimension-free auditing scheme is extended to provide an efficient dimension-free decision calibration algorithm in this setting.

**Strengths:**

This is a good paper; the proposed array of results are both insightful, creating previously unknown separations, and employ a variety of techniques that may be useful for follow-up work.

On the insights side, it is an important message that decision calibration can be done in a dimension free way; equally important, however, is the complementary message that the dimension-dependence can go away for smooth decision rules but not for the canonical nonsmooth one. This draws distant parallels to other, quite distinct, instances of smooth vs. nonsmooth issues in (regular) (multi)calibration, but is very much a decision-focused phenomenon here.

On the techniques side, here are a couple that I want to point out. First, extending from the Gopalan et al constructions, that applied to full calibration rather than the decision variant, is not fully trivial and required some careful constructions. Second, on the constructive side, constructing the dimension-free calibration algorithm using the auditing algorithm is also not as straightforward as it may seem; among all else, it required “implicit patching”, a simple but useful trick.

**Weaknesses:**

I believe this is a good-quality and innovative paper, and only have relatively smaller qualms with it.

First, the negative result doesn’t quite fully align with algorithmic impossibility of calibration itself (and I’m not convinced how difficult to solve this misalignment problem is in this context). Briefly (and as quickly mentioned in the corresponding section of the paper), the proof by covering shows that auditing for calibration is not possible to do without incurring a dimension-driven dependence. Given that all existing (decision) calibration algorithms depend on auditing of miscalibration, it is thus possible to conclude that the currently studied ubiquitous algorithmic template doesn’t allow for dimension-free calibration. However, there might ostensibly be an off-chance whereby decision calibration is possible to perform dimension-free by eschewing direct auditing. This off-chance possibility thus represents a gap that must be discussed more prominently; this includes a broader perspective — auditing/testing vs calibration tasks have both come up in the literature before e.g. for vanilla calibration, so are there any insights into how fundamental this gap is to bridge?

Secondly, the paper touches upon the following important consideration, which is to my knowledge fairly novel in the literature: That decision calibration can exhibit tractability separations depending on the nature of the decision rule. In this case, the very least that must be done is to align the presentation and the terminology surrounding the decision rules that are studied. In particular, given that (1) the hard best response rule comes up in the 0-temperature limit of quantal response, and (2) the bounds obtained in the upper bound side for quantal response have an explicit dependence on the inverse temperature beta, it would be clean to define and refer to decision rules throughout the paper in terms of that temperature. (Even on the current terminology level, the quantal response mapping is also called other names such as “smooth” etc. so that can be streamlined.)

In particular, I would consider the story about the influence of the temperature on the hardness of the problem somewhat fully addressed if the upper bounds involving beta were also accompanied by lower bounds in beta (ignoring other parameters besides beta); matching bounds may be a lot to ask for from the technical perspective, so any, even slow, lower-bounding dependence on beta would suffice to drive the above point home.

**Questions:**

My substantive questions in this case are limited to the issues in the Weaknesses section above. In addition to those, I have observed a fair amount of typos and grammar issues, which would be tedious to list here (but suffice it to say, they are prominent even in the abstract) --- so to improve the writing, these need to be fixed.

---

> ### Author Response · Authors · 2025-11-24
>
> Thank you for taking the time to review our submission! Please find our replies to your questions/comments below.
>
> **W1 regarding the algorithmic lower bound:**
>
> Thank you for raising this interesting question! Establishing algorithmic lower bounds in the (multi)calibration literature remains largely open. Even for the more standard notion of expected calibration error (ECE), to the best of our knowledge, prior work has not provided a clean algorithmic lower bound. One challenge is that a constant predictor that simply outputs the empirical mean is already sufficient to achieve decision calibration (and even the stronger notion of ECE). Thus, deriving a meaningful lower bound that rules out such constant predictors would require additional assumptions or constraints.
>
> The most relevant existing result is due to Gibbs and Tibshirani (2025) [1], who prove a lower bound for one-dimensional ECE under an additional multi-accuracy constraint over a family of singleton functions. Inspired by their result, one potential direction is to study lower bounds for learning a predictor that simultaneously achieves decision calibration and small squared loss. We view this as an interesting open problem for future work.
>
> **W2 regarding the lower bound with $\beta$:**
>
> Thank you for the great question! We have proved a lower bound showing that the sample complexity of auditing decision calibration is $\Omega(\tanh^2(c\beta)/\epsilon^2)$, for a constant $c$. The proof constructs two distributions that are statistically close: one is 0-decision calibrated and the other has decision calibration error $\epsilon$. Let $\mathcal{Y}=\{0,1\}$. For $\mathcal{D}_0$, set $\mathbb{E}[Y]=c$; for $\mathcal{D}_1$, set $\mathbb{E}[Y]=c+\epsilon/\tanh(\beta c)$. A standard Taylor expansion shows that $\mathrm{KL}(D_0\|D_1)=\Delta^2/(8 q_0(1-q_0)) + O(\Delta^3)$, where $\Delta=\epsilon/\tanh(\beta c)$ and $q_0=(c+1)/2$. Therefore, for $n$ i.i.d. samples, $\mathrm{KL}(D_0^{\otimes n}\|D_1^{\otimes n})=O(n\,\epsilon^2/\tanh^2(\beta c))$. By Pinsker’s inequality, the total variation distance is $O(\sqrt{n\,\epsilon^2/\tanh^2(\beta c)})$, which implies that achieving error below a constant requires $n>\Omega(\tanh^2(\beta c)/\epsilon^2)$.
>
> When $\beta$ is small, $\tanh(\beta c)\approx c\beta$, giving a quadratic dependence on $\beta$. However, when $\beta\to\infty$, $\tanh(\beta c)\le 1$. Although this is not the lower bound the reviewer speculated about, we believe this is mostly the correct behavior: any lower bound growing polynomially or even logarithmically in $\beta$ as $\beta\to\infty$ would imply that sample complexity becomes intractable in the hard-max limit, contradicting Zhao et al. (2021)’s positive result showing that decision calibration under strict best response is sample-efficient in finite-dimensional outcome spaces.
>
> Ideally, one might strengthen this by proving a lower bound of the form $\Omega(f(m,\beta)/\epsilon^2)$, where $f(m,\beta)\approx\beta^2$ as $\beta\to 0$ and $f(m,\beta)\to\mathrm{poly}(m)$ as $\beta\to\infty$. We believe it is an interesting open question to establish such an interpolation.
>
> **Q1 regarding the typos:** Thank you for pointing this out. We will correct the typos in the final version.
>
> References:
>
> [1] Gibbs, Isaac, and Ryan J. Tibshirani. "Sample-Efficient Omniprediction for Proper Losses." arXiv preprint arXiv:2510.12769 (2025).

---

### Official Review · Reviewer_Zo8a · 2025-11-04

**Soundness:** 3
**Presentation:** 2
**Contribution:** 3
**Rating:** 4
**Confidence:** 4

**Summary:**

Prior work on decision calibration requires that predictions are unbiased conditional on events relevant to action selection under a linear loss function. This paper extends the decision calibration framework to a setting where decision makers to have nonlinear loss functions.

**Strengths:**

- Most prior work in decision calibration, e.g. Zhao et al 2021, Sahoo et al 2021, focuses on linear losses, so this work makes a contribution to the area by considering nonlinear loss functions.

- This work proposes to approximate nonlinear loss functions using a feature mapping $\phi: \mathcal{Y} \rightarrow \mathcal{H}$, so that a nonlinear loss function $\ell(y, a)$ can be approximated by a linear loss function $\ell^{*}(\phi(y), a)$. It is quite common and natural to express a nonlinear function through a basis expansion.

- The main theoretical contributions include a lower bound for dimension-free decision calibration, a dimension-free sample complexity guarantee of decision calibration under nonlinear losses and quantal response, and improved sample complexity results for decision calibration.

**Weaknesses:**

- The introduction could be significantly strengthened by making clear why existing approaches for decision calibration fall short for nonlinear loss functions. In addition, it would also be helpful to emphasize the challenge that arises when moving from linear loss functions to nonlinear loss functions. Is decision calibration more difficult to achieve for nonlinear loss functions because we may require unbiasedness of higher-order moments of the outcome? Is this the motivation for embedding outcomes into a feature space?

- In a related vein, the link between nonlinear loss functions and achieving decision calibration for higher-dimensional outcomes could be made much more clear in the introduction. The abstract of the paper emphasizes moving from linear losses, which are commonly studied in decision calibration, to nonlinear loss functions. So, my initial interpretation was that this paper would focus on a loss functions that depend on higher-order moments of the outcome. However, the introduction very quickly jumps from a discussion of linear losses to a discussion of achieving decision calibration for high-dimensional outcome spaces.

- Since the authors propose to express a nonlinear function as a linear basis expansion, is it necessary to achieve some calibration notion for predictions of the potentially high-dimensional outcome $\phi(Y)$. I would expect the sample complexity of decision calibration to depend on the dimension of $\phi(Y)$ as in Zhao et al, 2021, the sample complexity depends on $C$ the number of classes.  It would helpful to clarify what's different/new about this analysis that allows us to sidestep the dependence on the dimension of $\phi(Y)$.

- It would be useful to clarify in the introduction why the authors consider under the quantal response, and what limitations are in considering the quantal response rather than just the best response rule.

**Questions:**

N/A

---

> ### Author Response · Authors · 2025-11-24
>
> Thank you for taking the time to review our submission! Please find our replies to your questions/comments below.
>
> **W1 & W2 regarding the introduction**:
>
> The reviewer asks why decision calibration is harder for non-linear losses and why this leads to high-dimensional outcome spaces.
> - Why existing methods fail for non-linear losses? On one hand, from a technical perspective, prior results on decision calibration (e.g., Zhao et al., 2021) rely crucially on the linearity of the loss function. Linearity ensures that best responses correspond to linear classification boundaries, which makes it possible to design algorithms whose sample complexity reduces to the Rademacher complexity of the class of best-response functions induced by linear losses. This complexity is polynomial in the dimension of the outcome space, enabling tractable algorithms for decision calibration. Without the linearity assumption, the original analysis cannot go through.
> On the other hand, decision calibration is conceptually useful because agents with linear utility functions best responding to the predictor incur no regret. However, for non-linear losses, such no-regret guarantees fail to hold even if the predictor is Bayes optimal, i.e., $p(x) = \mathbb{E}[y \mid x]$.
>
> - Why do we aim for dimension-free guarantees for high-dimensional feature/outcome spaces for non-linear losses? As we describe in the introduction, a natural approach for handling non-linear losses $l(a,y)$ is to linearize them via a feature map $\phi(y)$ such that $l(a,y) \approx \langle r, \phi(y) \rangle$. This mapping often requires the dimension $m$ of the feature space $\phi(y)$ to be high-dimensional or infinite (e.g., RKHS of universal kernel). Consequently, as we state in the introduction, standard decision calibration becomes computationally and statistically intractable when applied to these high-dimensional or even infinite-dimensional feature expansions $\phi$. Our contribution is specifically solving this "curse of dimensionality" to achieve sample complexity independent of $m$.
>
> Thank you for the suggestion. We have revised the introduction to clarify these points.
>
>
> **W3 regarding the intuition behind dimension-free guarantees**:
>
> Thank you for the question! As pointed out in the proof sketch of theorem 4.1,we sidestep the dependence on the dimension by considering the quantal response instead of the hard best response. Intuitively, the quantal response avoids sharp, discontinuous decision boundaries. The smoothness of the quantal response allows us to project the loss vectors in any high-dimensional feature space into 1-D space by a pseudometric:
> $$
> d(r_{l_1}, r_{l_2}) = \sqrt{\mathbb{E}[\langle r_{l_1} - r_{l_2}, p(X) \rangle^2]}
> $$
> The smoothness of the quantal response implies that if the pseudometric between two loss vectors is small, their decision rule is also close. This smoothness allows our analysis to rely on the projected 1D pseudo-metric, hiding the infinite-dimensional feature space $\phi(Y)$ from the auditing algorithm.
>
> **W4 regarding the quantal response**:
>
> Thank you for the question!
>
> - As pointed out in our paper, quantal response harks back to an extensive line of work on quantal choice theory in the economics literature. In contrast to best responses which assume perfectly rational agents who always choose the loss-minimizing action, quantal response accommodates noisy or boundedly rational decision making.
> - We adopt the smooth decision rule technically because we prove that dimension-free calibration is impossible under the standard deterministic best response. We establish this lower bound in Theorem 3.1, showing that deterministic rules require sample complexity scaling with $\sqrt{m}$, where $m$ is the outcome dimension.
> - Finally, we emphasize that our result for the quantal response model extends to any response function $\tilde{k} _{f,\ell}$ that maps estimated losses to a distribution over actions, as long as two conditions are satisfied: (i) the response function gives an approximate best response, and (ii) the induced action probabilities $\tilde{k} _{f,\ell}(x,a)$ are Lipschitz in the estimated losses. In all of our proofs, we rely only on these two properties and do not use any features specific to quantal best responses. We adopt the quantal response model in the paper primarily for simplicity, and because it is one of the most commonly used behavioral models for smoothed responses.

---

### Meta-Review · Area_Chair_zL4q · 2026-01-09

**Summary:**

Reviewers concerns covered most areas of the paper (introduction -- Zo8a --, terminology regarding its novel contributions -- zBGQ --, counter-intuitive part of some contributions -- zBGQ --, additional points to strenghten further the narrative -- F8eW, YoMP --). Reading the paper for myself, many of those concerns were fair and stated fairly.

**Reviewer Concerns:**

The authors made a fair treatment of all concerns in their answers. Special mention for the analytical treatment of W2 for zBGQ.

**Reviewer Scores:**

With the exception of review Zo8a, all others were already having substantial polarity in their evaluation of the paper so would have probably not changed further. Eventually Zo8a would have been slightly bumped up (review with a low absolute margin on acceptance).

---

### Decision · Program_Chairs · 2026-01-26

Accept (Poster)